# scCobra allows contrastive cell embedding learning with domain adaptation for single cell data integration and harmonization

Bowen Zhao[1,2,3], Kailu Song[2,4], Dong-Qing Wei [1], Yi Xiong [1] ✉ & Jun Ding [2,3,4,5,6] ✉

The rapid advancement of single-cell technologies has created an urgent need for effective methods to integrate and harmonize single-cell data. Technical and biological variations across studies complicate data integration, while conventional tools often struggle with reliance on gene expression distribution assumptions and over-correction. Here, we present scCobra, a deep generative neural network designed to overcome these challenges through contrastive learning with domain adaptation. scCobra effectively mitigates batch effects, minimizes over-correction, and ensures biologically meaningful data integration without assuming specific gene expression distributions. It enables online label transfer across datasets with batch effects, allowing continuous integration of new data without retraining. Additionally, scCobra supports batch effect simulation, advanced multi-omic integration, and scalable processing of large datasets. By integrating and harmonizing datasets from similar studies, scCobra expands the available data for investigating specific biological problems, improving cross-study comparability, and revealing insights that may be obscured in isolated datasets.

Single-cell genomics technology significantly advances the exploration of complex and heterogeneous biological systems by providing high-resolution measurements of cellular states. Over recent years, the proliferation of single-cell maps detailing diverse cell types from various tissues across numerous organisms, including humans and mice, has been observed[1–5]. These comprehensive single-cell atlases are often compiled across multiple batches, necessitated by the limitations of current single-cell sequencing technologies, experimental constraints such as budget or sample availability, or the pooling of datasets from different laboratories. Additionally, the use of various single-cell sequencing platforms, like 10X Genomics[6], micro-well[3], and Smart-seq[7], contributes to systematic technical variations known as "batch effects"[8–11] between single-cell measurements across different batches. To derive a more accurate and deeper understanding of biological insights from those single-cell maps, it is imperative to mitigate batch effects among single-cell datasets and ensure their integration and harmonization[12] for a comprehensive analysis.

To mitigate the impact of such batch effects on downstream single-cell data analytics, efficient computational methods are essential to integrate single-cell data from different batches and remove the batch effects. As of this date, dozens of single-cell RNA-seq (scRNA-seq) data integration methods have been developed. The most widely used methods include Seurat[13], Harmony[14], Scanorama[15], scVI[16], scDML[17], and scDREAMER[18]. Seurat uses canonical correlation analysis to project the cells into a reduced latent space to identify correlations across datasets. The mutual nearest neighbors (MNNs) are then identified in the reduced latent space and serve as "anchors" to align the data from different batches. In the PCA reduced space, Harmony iteratively removes the present batch effects. In each iteration, the method clusters cells from different batches while maximizing the diversity of batches within each cluster and then calculates a correction factor for each cell to be integrated. Another method, Scanorama also searches for MNNs in dimensionality-reduced space and uses them to guide batch correction. Unlike Seurat, which searches similar cells across batch pairs to compute the correction, Scanorama searches across all batches and calculates the dataset merging priority based on the matching cell percentage in each batch. scVI specifically models the gene expression variance induced by the library size difference and batch effects, which will be added

¹State Key Laboratory of Microbial Metabolism, School of Life Sciences and Biotechnology, Shanghai Jiao Tong University, Shanghai, China. ²Meakins-Christie Laboratories, Department of Medicine, McGill University Health Centre, Montreal, QC, Canada. ³Division of Experimental Medicine, Department of Medicine, McGill University, Montreal, QC, Canada. ⁴Quantitative Life Sciences, McGill University, Montreal, QC, Canada. ⁵School of Computer Science, McGill University, Montreal, QC, Canada. ⁶Mila-Quebec AI Institute, Montreal, QC, Canada. ✉e-mail: xiongyi@sjtu.edu.cn; jun.ding@mcgill.ca

as additional modules to a variational autoencoder (VAE) that learns the cell embeddings by minimizing the cell gene expression reconstruction error, thereby separating biological signals from batch effects. scDML first identifies MNNs and then applies a triplet-based approach to minimize the distance between positive cells and anchor cells while maximizing the distance between negative cells and anchor cells, thereby achieving batch correction. scDREAMER[18] combines the concepts of generative adversarial networks (GANs) and VAEs to eliminate batch effects in scRNA-seq data.

Although existing methods have demonstrated success in integrating scRNA-seq datasets from different batches and even distinct sequencing platforms[14], various problems and limitations remain, leaving room for further improvement. MNNs-based models find similar cells between batches and calculate the correction factor to mitigate the difference across batches. Therefore, this category of methods often over-corrects the batch effects since they tend to ignore the potential inherent difference associated with different batches, which could be problematic in many applications[19]. Specifically, depending on specific experimental scenarios, the cellular states of even the same cell types across different experimental batches could differ, which is quite common in many disease studies, in which the same cell type (e.g., Macrophages) could be significantly different between healthy and disease patients[20]. Therefore, ignoring those biological differences between batches may cripple the downstream scRNA-seq data analytics. Deep learning-based methods such as scVI and scDREAMER (both methods are based on autoencoders) learn the reduced cell embeddings in the latent space by reconstructing the input gene expression. It often assumes that the gene expression (i.e., raw count) follows a negative binomial distribution (or zero-inflated negative binomial)[21,22], but it is challenging to find a universal gene expression distribution for various scRNA-seq datasets from different studies and by different sequencing platforms[23,24]. A proper selection of the gene expression distribution will determine the reconstruction loss and dramatically influence the cell embedding learnings, which is critical for the downstream batch correction[25]. Finally, certain methods and their original implementations are challenging to apply in the original feature space[8,14,17,18], leading to a loss of model interpretability and making it difficult to understand the differences between various cellular states at the genetic level.

Eliminating batch effects not only facilitates the integration of multiple datasets, but also augments downstream tasks such as label transfer, batch generation, and multimodal batch correction. For label-transfer tasks, some methods, such as Seurat and MNN, require mapping new data onto a reference atlas for cell type annotation. However, this process is computationally intensive due to the large number of cells in the reference atlas. Moreover, these methods necessitate model retraining whenever new data becomes available, limiting the efficient utilization of pre-trained models[13,14,16,26]. Besides, the evaluation of current batch correction methods heavily relies on the simulation datasets[27,28]. However, there remains a discrepancy between the simulated batch-affected scRNA-seq data and its real scRNA-seq data counterpart[29]. This can easily result in inaccurate evaluations and benchmarking of batch correction methods' performance. In addition, integrating different omics data types offers a more comprehensive perspective of cellular states. While some methods have shown effectiveness in reducing batch effects in scRNA-seq data[12,30], their ability to incorporate single-cell Assay for Transposase-Accessible Chromatin sequencing (scATAC-seq) data and tackle batch effects across multi-omic datasets remains limited. Furthermore, many current multi-omics integration methods are performed in latent space[31-35], which is not conducive to using the mature scRNA-seq analysis workflows and makes it difficult to understand multimodal data at the gene level. Additionally, integrating spatial omics and scRNA-seq data poses significant challenges, primarily due to the limited number of detectable genes in spatial omics data.

To address the above limitations, here we introduce a deep neural network framework scCobra, which employs contrastive learning at both cellular and cluster levels, VAE, and GANs to integrate single-cell data across varying batches. scCobra distinguishes itself through its comprehensive approach to data integration and harmonization across single-cell studies. Its proficiency spans a range of critical functions, from effectively correcting batch effects in scRNA-seq datasets to handling multi-omic data variations and generating meticulously detailed scRNA-seq datasets for rigorous benchmarking.

The capabilities of scCobra in batch correction have been validated across various tasks, including scRNA-seq, spatial omic, and single-cell multi-omic. Through both simulation and analyses of real disease datasets, scCobra has been shown to effectively minimize the risk of over-correction in batch adjustments. Beyond batch correction, scCobra enables a range of downstream data harmonization analyses, such as online cell label propagation and the creation of benchmark batch-affected scRNA-seq datasets enriched. The introduction of scCobra equips the single-cell genomics research community with a tool designed to tackle batch effects, enhancing the integration and harmonization of relevant single-cell datasets and aiding in the generation of precise and reliable biological insights.

## Results
### scCobra model overview
scCobra is a deep neural network framework that employs the contrastive VAE–GAN architecture, designed for the integration and harmonization of scRNA-seq data across multiple batches (Fig. 1). Using an encoder, scCobra maps the scRNA-seq data across different batches into a latent low-dimensional space. A decoder with a Domain-Specific Batch Normalization (DSBN) layer then reconstructs the original inputs with batch effect from the latent representations.

The reconstruction objective of scCobra has three primary facets. The first is an adversarial training component, ensuring that the reconstructed data closely mirrors the original. The second part is the reconstruction loss, which ensures that the batch information is accurately reconstructed. The third is a contrastive learning component, split between cell-level and cluster-level mechanisms. At the cell level, it ensures maximal similarity between original cells (view1) and their reconstructed counterparts (view2) in the latent space. Meanwhile, at the cluster level, it requires that the clustering representations obtained from the original cells (view1) remain consistent with the corresponding clustering representations derived from the reconstructed cells (view2). A domain discriminator is also incorporated to remove batch information from the latent embedding $\mu$. Our method can also deliver batch correction in the original input space by reconstructing all input cells from the batch-corrected latent space via the same BN layer (e.g., BN1). The versatility of scCobra's batch-corrected outputs, effective in both latent and original spaces while also reducing the risk of over-correction, makes it highly applicable for a variety of downstream tasks. These include batch correction, multi-omic data integration, online label transfer, and the simulation of scRNA-seq datasets with batch effects. Detailed description of the scCobra model is available in the "Methods" section.

### scCobra outperforms benchmarked methods in batch effect correction
We employed the scCobra method on several datasets: a complex human lung atlas dataset[27,36], an immune cell bone marrow dataset[37-41], and a human pancreas dataset encompassing data from different sequencing platforms[42-46]. To benchmark the performance and evaluate the effectiveness of scCobra, we also utilized several state-of-the-art scRNA-seq batch correction methods, namely Seurat, Harmony, Scanorama, scVI, scDML, and scDREAMER, on these datasets for comparative analysis.

The human lung atlas dataset[27,36], which includes 16 batches, 17 cell types, and over 32,000 cells, provides a highly challenging benchmark for batch correction methods. UMAP visualizations (Fig. 2a, b) show that scCobra, together with scVI, achieved the best performance in distinguishing cell types and integrating batches, while other methods exhibited notable limitations. Seurat struggled to separate multiple cell types. Harmony mixed Type 2 and Basal 2 cells, as well as "Neutrophil_CD14_high" cells with Macrophages and Scanorama, while differentiating Basal 1 and Basal 2 cells, failed to integrate Macrophages and Type 2 cells across batches. Similarly, scDML and scDREAMER encountered challenges in distinguishing specific cell types such as "Neutrophil_CD14_high" cells and

Macrophages. Quantitative results (Fig. 2c) further validate scCobra's superior performance. It achieved the highest Adjusted Rand Index (ARI), Normalized Mutual Information (NMI), and CellType ASW scores, demonstrating its ability to preserve biological signals while effectively integrating batches. In terms of Graph Connectivity, scCobra performed comparably to scVI and scDREAMER, whereas Scanorama attained the highest Batch ASW but exhibited under-correction tendencies in UMAP visualizations (Fig. 2a, b).

The inherent complexity of the immune dataset[27], featuring over 33,000 cells across 12,303 genes and spread over 10 batches and 16 distinctive cell types, provides an ideal canvas to assess the robustness of various integration methods in challenging scenarios. Upon examining the UMAP visualization, it's evident that several methods face difficulties (see Supplementary Fig. S1a). Scanorama, despite its merits, showcases its limitations in such intricate environments. It grapples with the unification of cells from varied datasets, especially in seamlessly integrating subsets like the $CD4+$ T

cells and $CD14+$ Monocytes. scVI, although effective in batch integration, struggles to maintain clear cell type boundaries, as demonstrated by its suboptimal integration of $CD20+$ B cells and $CD4+$ T cells. This blurring of cell types raises concerns about the reliability of downstream analyses. Similarly, scDML exhibits notable classification errors, merging $CD4+$ T cells with monocyte-derived dendritic cells into a single cluster and incorrectly splitting $CD20+$ B cells into two distinct clusters, further underscoring its limitations in accurately resolving cell populations. The benchmark metrics not only underscore its solid batch correction capabilities but also laud its precision in clustering that echoes the ground-truth labels, particularly in such intricate datasets (as illustrated in Fig. 2d).

Batch effects arising from variations in single-cell RNA sequencing platforms for pancreas data pose a significant challenge for data integration. Given this, we further assessed scCobra's capability to harmonize scRNA-seq data from varying sequencing platforms using a pancreas scRNA-seq dataset[27]. This dataset encompasses cells derived from six distinct

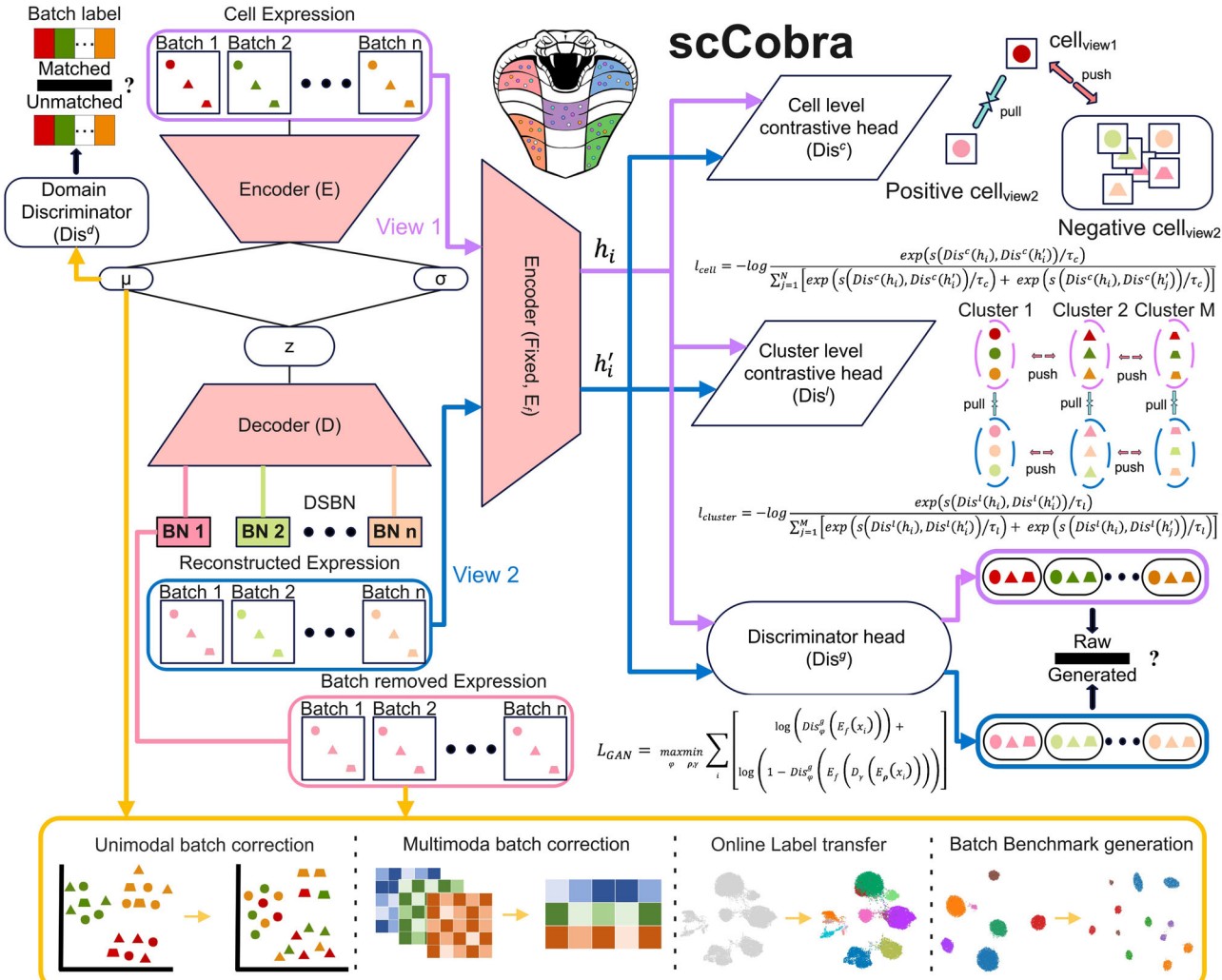

**Fig. 1 | scCobra method overview.** scCobra integrates a contrastive variational autoencoder with a generative adversarial network architecture to process multiple scRNA-seq data batches. This model features an encoder that consolidates inputs from various batches into a unified low-dimensional latent space. A decoder, enhanced with a Domain-Specific Batch Normalization (DSBN) layer, then reconstructs the original batch data. The DSBN layer is specifically designed to recognize and adjust for batch effects, enabling accurate reconstruction of batch-influenced data from the latent space. The reconstruction aim of scCobra is split into three primary objectives. The first is adversarial training, focused on increasing the resemblance between the reconstructed and original data. The second part is the reconstruction loss, which ensures that the batch information is accurately

reconstructed. The third leverages contrastive learning at both cell and cluster levels to ensure precision in data replication and category consistency between original and generated cells. A domain discriminator further refines the model by mitigating batch-specific signals from the latent embeddings ($\mu$), enhancing the purity of the latent space representation. The dual outputs of scCobra, from both latent ($\mu$) and original space (Batch-removed Expression, all batches mapped to BN1), support a broad spectrum of downstream applications. These include but are not limited to unimodal batch correction, multimodal batch correction, label transfer, and the generation of benchmark datasets, demonstrating scCobra's adaptability and utility in single-cell data analysis.

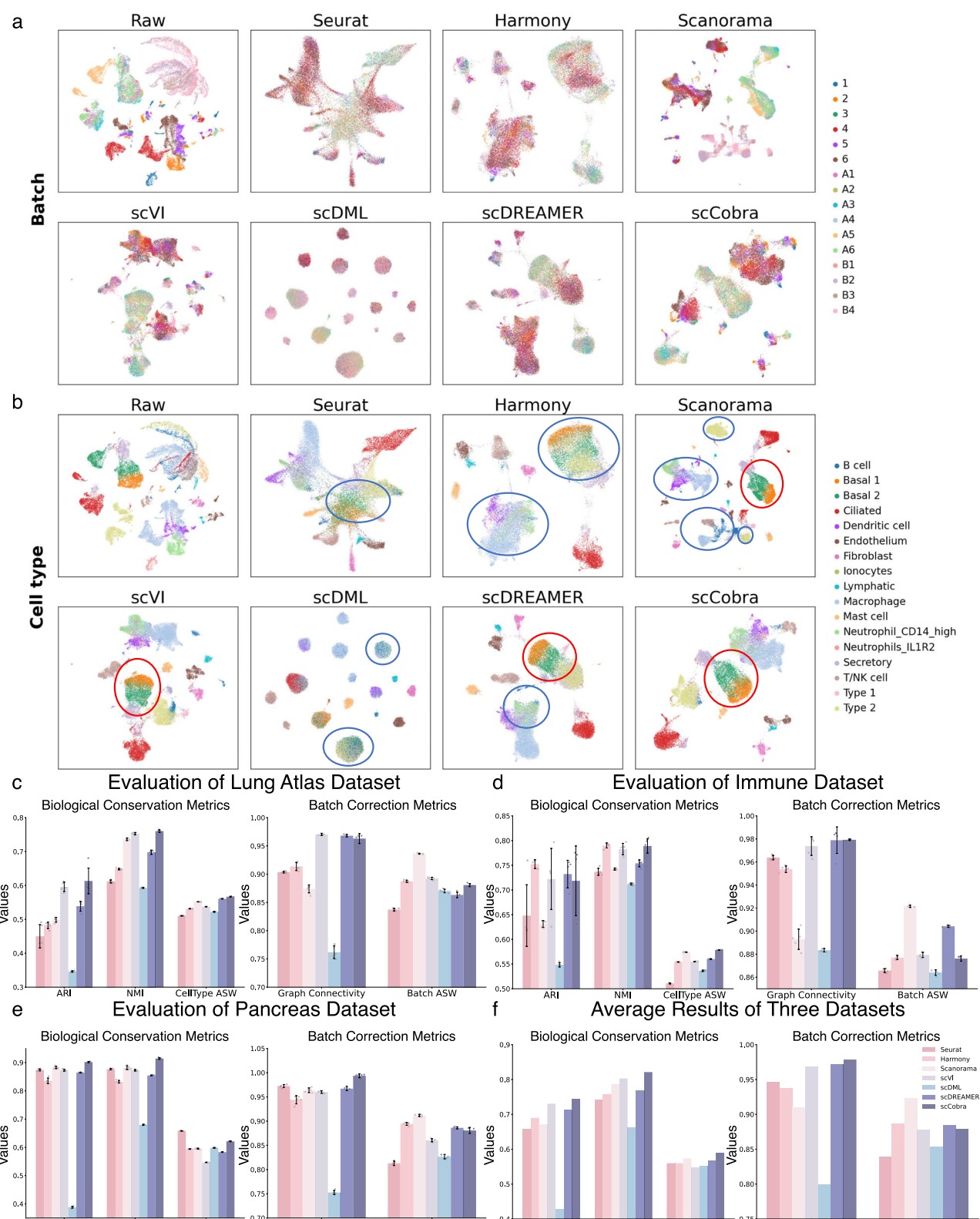

sequencing platforms (CEL-seq[47], CEL-seq2[48], Fluidigm c1[49], InDrop[50], and Smart-seq2[51]) spread across eight batches. The UMAP visualization (Supplementary Fig. S1b) provides insightful comparative results. Scanorama exhibited a limited correction capacity, evident from the discernible batch effects and the suboptimal integration of Alpha and Beta cells. Meanwhile, scDML failed to distinguish alpha cells from alpha_er cells and split acinar cells into two separate clusters, compromising the clarity needed to

distinguish these cell types without specific markers. Contrastingly, scCobra and Seurat emerged as the methods adeptly clustering alpha_er cells without confusion, whereas other tested methods showed a propensity to conflate alpha cells with alpha_er cells. In addition, scCobra's UMAP plots were notably distinct, enabling clear delineation between various cell types, a characteristic immensely valuable for accurate clustering. Benchmark evaluations highlight scCobra's robust performance compared to other

**Fig. 2 | scCobra demonstrates superior batch correction performance over state-of-the-art methods. a, b** UMAP visualizations comparing scCobra with benchmarked methods (Seurat, Harmony, Scanorama, scVI, scDML, and scDREAMER) on the human lung atlas dataset. **a** The batch correction results, with different batches distinguished by color, while **b** cell type aggregation, with each cell type assigned a unique color. Regions of superior performance are highlighted with red circles, indicating areas where scCobra or the benchmarked methods excel, whereas blue circles denote areas requiring improvement. **c–e** Quantitative results of scCobra and the benchmarked methods across different datasets (*n* = 5 independent experiments). We provide two types of evaluation metrics. Biological conservation metrics include ARI, NMI, and CellType ASW, with higher values indicating better retention of biological signals during batch effect removal. Batch correction metrics include Graph Connectivity and Batch ASW, where higher values signify better mixing of cells from different batches and more effective removal of batch effects. This panel differentiates the performance of each method using color coding, providing a clear comparison of their effectiveness in addressing batch effects and enhancing data comparability. **f** Average quantitative results of scCobra and benchmark methods across various datasets demonstrate the robust batch correction performance of scCobra. Error bars represent the standard deviation of the evaluated metrics across datasets.

methods. It achieved the highest scores in ARI, NMI, and Graph Connectivity (Fig. 2e), while maintaining competitive performance in CellType ASW and Batch ASW metrics. Collectively, these outcomes attest to scCobra's prowess in eradicating batch effects from scRNA-seq data sourced from diverse platforms.

As summarized in Fig. 2f, scCobra achieves the highest average performance across ARI, NMI, CellType ASW, and Graph Connectivity, highlighting its robust balance between biological information retention and batch effect correction. Specifically, scCobra demonstrates exceptional performance in ARI, NMI, Graph Connectivity, and CellType ASW, underscoring its ability to preserve biological structure. At the same time, its competitive Graph Connectivity and Batch ASW scores ensure effective batch effect removal and seamless data integration. These results position scCobra as a leading method for preserving biological fidelity and addressing batch effects, particularly in the context of complex scRNA-seq datasets. We also provide batch-correction results on the simulation dataset (Supplementary Fig. S2). However, due to the dataset's simplicity, which does not adequately capture the complexity of real single-cell datasets, these simulation results were excluded from the benchmarking analysis.

We further benchmarked scCobra's performance against other methods in another scenario by comparing the number of differentially expressed genes (DEGs) between samples under the exact same condition, where no significant biological differences are expected. Using raw data from two healthy samples in the liver cancer dataset[52], we identified 470 DEGs, reflecting substantial batch noise. After batch correction, scCobra again demonstrated superior performance by reducing the number of DEGs to 30, compared to scVI (66) and Harmony (40). These results further highlight scCobra's robustness in effectively removing batch noise more efficiently than other methods, while preserving meaningful biological signals (Supplementary Fig. S3).

## scCobra minimizes over-correction risks in batch correction

The goal of batch correction is to minimize these technical variations so that the true biological variations across samples can be accurately identified and analyzed. However, over-correction occurs when the batch correction algorithm is too aggressive, leading to the loss of relevant biological information, which is a prevalent challenge among existing methods and often hinders the adoption of batch correction in practical applications[19,26,53]. To showcase that scCobra can minimize the over-correction risks in comparison to its counterparts, here we applied the methods to a simulated dataset (human immune dataset[39,41,54]) and two real scRNA-seq datasets (Liver cancer[52], COVID-19[55]).

First, we adjusted the gene expression levels of *CD4*+ T cells in batch "10X" from a human immune dataset by modifying standardized expression values for genes in the "Defense response to Virus" Gene Ontology Biological Process (GOBP) term (Supplementary Fig. S4). This was performed to simulate the gene expression changes in *CD4*+ T cells following viral infection, designating this modified group as "perturbed *CD4*+ T cells". Theoretically, after batch correction, these two populations should remain distinct with minimal overlap. Greater overlap between these populations indicates a higher degree of over-correction. To evaluate this, we applied batch correction tools such as Seurat, Harmony, Scanorama, scVI, scDML, scDREAMER, and scCobra, and generated UMAP visualizations (Fig. 3a). The visualizations revealed that Seurat, Harmony, scDML,

and scDREAMER failed to distinguish between perturbed and unperturbed *CD4*+ T cells. While Scanorama and scVI, were both able to differentiate between perturbed and unperturbed *CD4*+ T cells similarly to scCobra, both methods exhibited limitations. Scanorama showed a tendency toward under-correction, with cells of the same type, such as *CD4*+ T cells, failing to integrate cohesively and instead appearing scattered across the UMAP space. This lack of proper integration extended to other cell types, including *CD8*+ T cells and *CD14*+ monocytes, underscoring the limitations of Scanorama in achieving robust and consistent data integration. Similarly, while scVI was able to separate perturbed and normal *CD4*+ T cells, the separation was not as clear or consistent as that achieved by scCobra. Additionally, scVI displayed limitations in maintaining integration within the same cell type across batches, leading to suboptimal clustering in certain regions of the UMAP space. In contrast, scCobra successfully separated perturbed and normal *CD4*+ T cells with minimal overlap while maintaining cohesive integration of cells within the same cell type, outperforming both Scanorama and scVI.

We further illustrate the separation of scCobra-corrected *CD4*+ T cells and perturbed *CD4*+ T cells using marker genes shown in Fig. 3b. The results reveal a clear and well-defined boundary between these two cellular states. This separation demonstrates that scCobra effectively minimizes over-correction risk by preserving the biologically meaningful differences between the perturbed and normal cells. Furthermore, marker genes such as *TLR9*, *HSP90AA1*, and *IL27*—associated with the "Defense response to virus" GOBP term—served as robust differentiators, highlighting scCobra's ability to maintain critical biological variation after batch correction.

In addition to the above visualizations, we further systemically quantified the extent of over-correction for each method using the over-correction score[26] (Fig. 3c) to demonstrate the superiority of scCobra in minimizing over-correction risk, with higher scores indicating a greater risk of over-correction and values closer to 0 reflecting minimal risk. When evaluating the dataset containing all cell types, scDML exhibited the highest degree of over-correction, followed by Seurat, Harmony, and scDREAMER, which demonstrated varying levels of over-correction. Among the methods evaluated, scCobra showed a relatively low over-correction score (0.043), outperforming scVI (0.086) and significantly improving over methods like scDML. Although Scanorama reported the lowest over-correction score (0.012), its UMAP visualization (Fig. 3a) revealed a tendency toward under-correction, as it failed to cohesively integrate cells of the same cell type. Focusing only on the subset of *CD4*+ T cells and perturbed *CD4*+ T cells, we also independently calculated over-correction scores. Seurat, Harmony, and scDREAMER demonstrated over-correction scores exceeding 0.35, while scVI (0.158) and scDML (0.278) also exhibited higher scores compared to scCobra, which achieved a notably lower score of 0.038. Although Scanorama again reported the lowest score (0.0), its under-correction tendency, as seen in the UMAP visualization (Fig. 3a), compromised the biological integrity of the data.

To ensure that our conclusions regarding over-correction risk are not confined to a specific cell type and perturbation (i.e., a specific GO term that we have chosen), we extended our perturbed simulation to include another cell type (i.e., *CD14*+ monocytes) and perturbation (e.g., "Reactome influenza life cycle") (Supplementary Figs. S5 and S6). Similarly, scCobra successfully differentiated perturbed *CD14*+ monocytes and unperturbed counterparts, whereas Seurat, Harmony, scVI, scDML, and scDREAMER

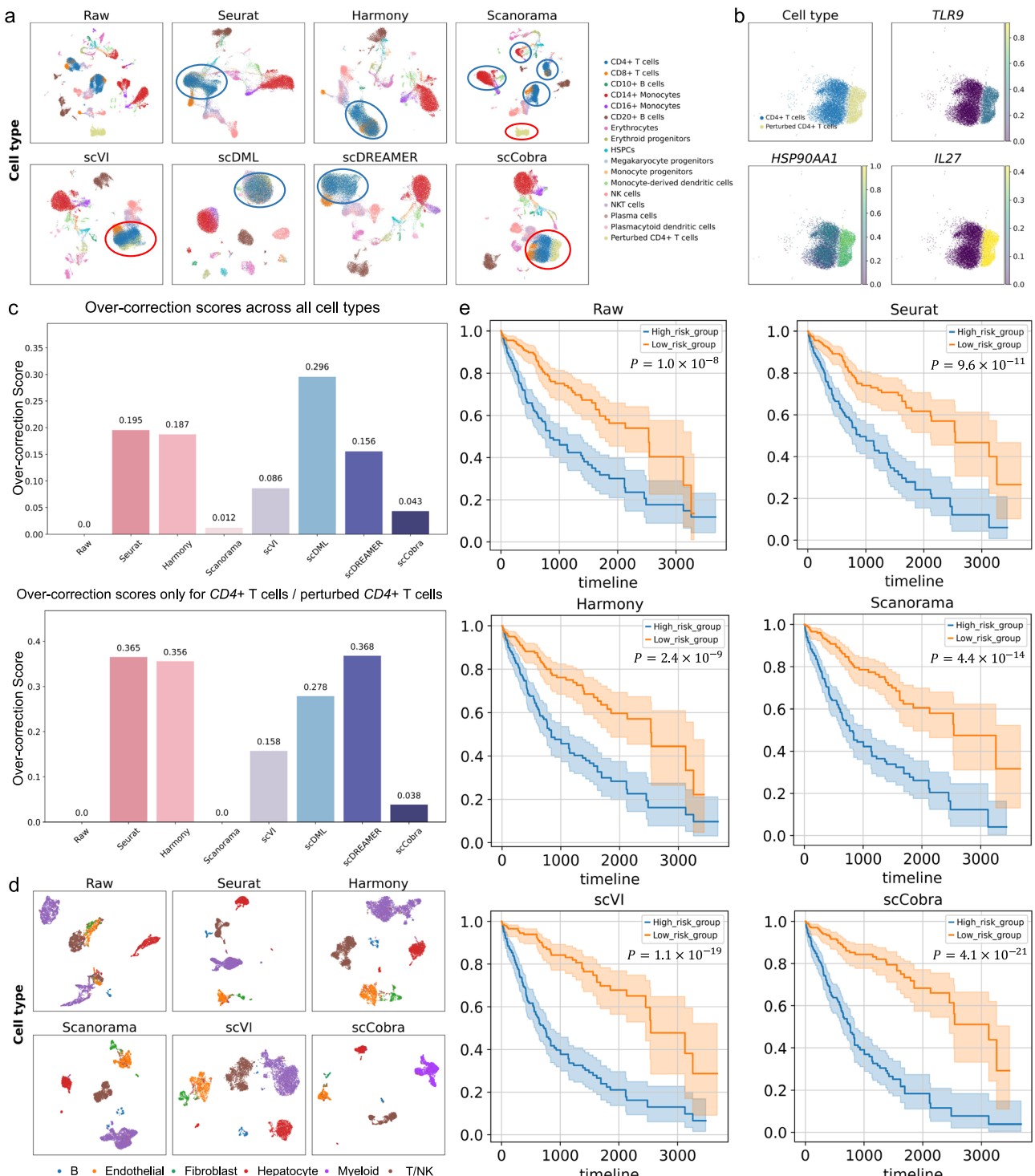

**Fig. 3 | scCobra can minimize over-correction risk. a** Batch correction methods (Seurat, Scanorama, Harmony, scVI, scDML, scDREAMER, and scCobra) were evaluated using a simulated immune dataset with an implanted biological signal introduced by perturbing genes associated with the "Defense response to virus" Gene Ontology Biological Process (GOBP) term. The UMAP visualizations in the figure are color-coded by cell type, illustrating each method's ability to preserve biological distinctions while mitigating batch effects. **b** The enlarged UMAP plots clearly indicate that scCobra separates normal from perturbed *CD4+* T cells, highlighted by specific markers *IL27*, *HSP90AA1*, and *TLR9*. **c** Quantitative over-correction analysis on all cell types and specifically for *CD4+* T cells and perturbed *CD4+* T cells

using normalized over-correction score. Higher values indicate a greater degree of over-correction, whereas values closer to zero reflect minimal over-correction. **d** The UMAP visualizations of the liver cancer dataset, integrated using different methods and colored by cell type, demonstrate the batch correction capabilities of each approach. **e** Survival analysis results using differential gene sets obtained from corrected gene expression with scCobra and benchmarking methods. Blue represents high-risk groups, and orange represents low-risk groups, with the *x*-axis indicating survival days and the *y*-axis representing the survival proportion. The statistical *P* values were determined by the two-tailed log-rank sum test.

failed. Additionally, scCobra consistently exhibited lower over-correction scores compared to these methods, further validating its ability to reduce the risk of masking true biological variation during batch correction.

To further elucidate the impact of batch noise and highlight the necessity of minimizing over-correction risk, we analyzed the liver cancer dataset[52]. After applying various batch correction methods, including Seurat, Harmony, Scanorama, scVI, and scCobra, UMAP visualizations confirmed that batch effects between the two samples had been effectively removed (Fig. 3d). However, while successful batch correction reduces technical variability, overly aggressive correction risks eliminating meaningful biological signals. To assess this, we performed survival analysis based on sets of DEGs identified from raw and batch-corrected data (Fig. 3e). Among all methods, scCobra achieved the best performance, producing the most robust survival-adjusted $P$ (e.g., survival-adjusted $P = 4.1 \times 10^{-21}$). In contrast, raw data ($1.0 \times 10^{-8}$), Seurat ($9.6 \times 10^{-11}$), Harmony ($2.4 \times 10^{-9}$), Scanorama ($4.4 \times 10^{-14}$) and scVI ($1.1 \times 10^{-19}$) yielded weaker results, indicating a loss of crucial biological information likely caused by over-correction. This highlights the importance of balancing batch effect removal with the preservation of genuine biological signals. To further validate the robustness of our findings, we performed differential gene analysis and survival analyses across varying thresholds, consistently confirming the reliability of scCobra's performance (Supplementary Fig. S7).

In addition, we applied Seurat, Harmony, Scanorama, scVI, and scCobra to correct batch effects between diseased and healthy samples in the COVID-19 dataset[55]. Using the biologically relevant GOBP term "Defense response to virus"[55] as an example, enrichment analysis revealed that scCobra preserved this critical biological signal, whereas other methods failed to retain it after batch correction (Supplementary Fig. S8). This finding exemplifies the risk of over-correction associated with other methods.

Our results demonstrate that scCobra effectively reduces the risk of over-correction during batch correction and is more capable of uncovering true biological signals compared to using raw data or other correction methods.

## scCobra enables multi-omic batch correction

While many contemporary methods target the correction of batch effects in scRNA-seq data exclusively, scCobra stands out due to its versatility, addressing batch effects across a spectrum of single-cell data types, not just specifically applied to scRNA-seq. Here, we utilized scCobra to harmonize scRNA-seq and scATAC-seq datasets. These datasets, provided as example data by 10X Genomics through Cell Ranger, originate from human peripheral blood mononuclear cells. We transformed the scATAC-seq peak matrix into a gene activity matrix using MAESTRO[56] and treated it like a special "scRNA-seq batch", streamlining the integration of single-cell multi-omic data into a batch correction task. However, even after converting the scATAC-seq peak matrix into a gene activity matrix, the modality differences with scRNA-seq data remained substantial. scCobra helped mitigate this issue, the batch-corrected fusion of the two omics data showcased a remarkable alignment: not only were cells of congruent types grouped, but those sharing analogous developmental arcs also neighbored each other. For instance, B cell progenitors aligned with pre-B cells, and *CD14+* Monocytes nestled alongside *CD16+* Monocytes (Fig. 4a). From the perspective of quantitative metrics, we used the Batch Entropy Mixing Score (Batch mixing score) and the Batch ASW score to quantify batch correction. Harmony, Scanorama and scDREAMER struggle to handle multi-omic data with significant modality differences, whereas scCobra, scVI, and Seurat are all capable of effectively integrating multi-omic data, methods in the upper right corner exhibit enhanced batch integration capabilities and more effectively eliminate batch effects (Fig. 4b). We further calculated three important biological conservation metrics: ARI, NMI, and CellType ASW, and summed them up. The results demonstrated that scCobra and Seurat effectively remove batch effects while retaining the most biological information (Fig. 4c).

To further enhance its applicability in data integration and harmonization, we combined scRNA-seq data with single-cell spatial MERFISH data[57] from the mouse hypothalamus region[57] (Fig. 4d). Due to the relatively fewer genes detected by MERFISH compared to scRNA-seq, it is challenging to integrate such data using a limited set of features. We selected genes expressed in both MERFISH and scRNA-seq data as inputs for testing. The results indicate that scCobra effectively integrated MERFISH data with scRNA-seq data, for instance, positioning newly formed OD cells (oligodendrocytes) adjacent to OD Immature cells. We used the same quantitative strategy employed for evaluating multi-omic datasets to assess the performance of scCobra and benchmark methods on spatial MERFISH data. From the perspective of batch correction, scDREAMER struggles with handling batch effects between scRNA-seq and MERFISH data. Seurat, Harmony, Scanorama, scDML, and scVI have similar performances, whereas scCobra demonstrates the best performance in handling batch effects (Fig. 4e). In terms of biological conservation metrics, scVI and Scanorama preserve a similar level of biological specificity. Seurat, Harmony, scDML, and scDREAMER perform better than the first two methods, while scCobra significantly outperforms all the competing methods (Fig. 4f).

## scCobra enables the simulation of scRNA-seq data with batch effects

While many scRNA-seq batch correction methods rely on integrating multiple datasets for performance evaluation, inherent variations in annotation standards across these datasets can induce biases, Some methods, such as Splatter[28], can simulate batch effects by setting different random seeds, but they cannot be applied to simulated data based on a real dataset, which potentially limits the authenticity of data and its application in benchmarked batch correction methods. Addressing this challenge, scCobra can simulate batch effects with a ground-truth reference based on an existing real scRNA-seq dataset, ensuring an unbiased evaluation of batch correction techniques. Specifically, for a given batch "n", scCobra first secures an embedding devoid of batch information using a pre-trained encoder. Thereafter, with its trained decoder and DSBN module, scCobra can simulate distinct batches from one original batch via different BN layers for decoding from the latent space (Fig. 5a). This approach guarantees uniform cell type annotations across the simulated batches, removing potential annotation inconsistencies.

To illustrate this functionality and the process of generation, we created a simulated scRNA-seq dataset that exhibits batch effects, derived from a consistent set of cells annotated with their cell types, thereby establishing a clear ground truth. This approach was aimed at mirroring real-world batch variations while maintaining precise biological annotations for subsequent validation. Initially, we extracted three batches, "indrop1", "smartseq2", and "celseq" from the pancreas dataset and addressed the batch effects present in this subset using scCobra, evaluating the correction's effectiveness through visual analysis with UMAP projections (Fig. 5b upper panel). Subsequently, scCobra demonstrated its versatility by generating simulated batches for comparative analysis. Specifically, it replicated cells from Batch "indrop1" and distributed them across the original Batch "indrop1" as well as two artificially created batches, "smartseq2" and "celseq."

In the visualizations produced by Batch UMAP, we employed a color-coding scheme where blue indicated cells from Batch "indrop1", orange for "smartseq2" and green for "celseq", facilitating an intuitive understanding of batch origins. Additionally, the cell type UMAP visualization was leveraged to distinctly categorize the cell types (Fig. 5b lower panel). Building on this foundation, we broadened our analysis to include the application of scCobra, Seurat, Harmony, Scanorama, scVI, scDML, and scDREAMER to this intentionally simulated dataset, aiming to scrutinize the batch integration efficacy of each method. All these methodologies proficiently facilitated the amalgamation of cell populations from both batches, as clearly depicted in the cell type UMAP, where cells of identical types were aggregated (Fig. 5c), with quantitative results further supporting the performance of these methods (see Supplementary Fig. S9). Similarly, we extracted two batches

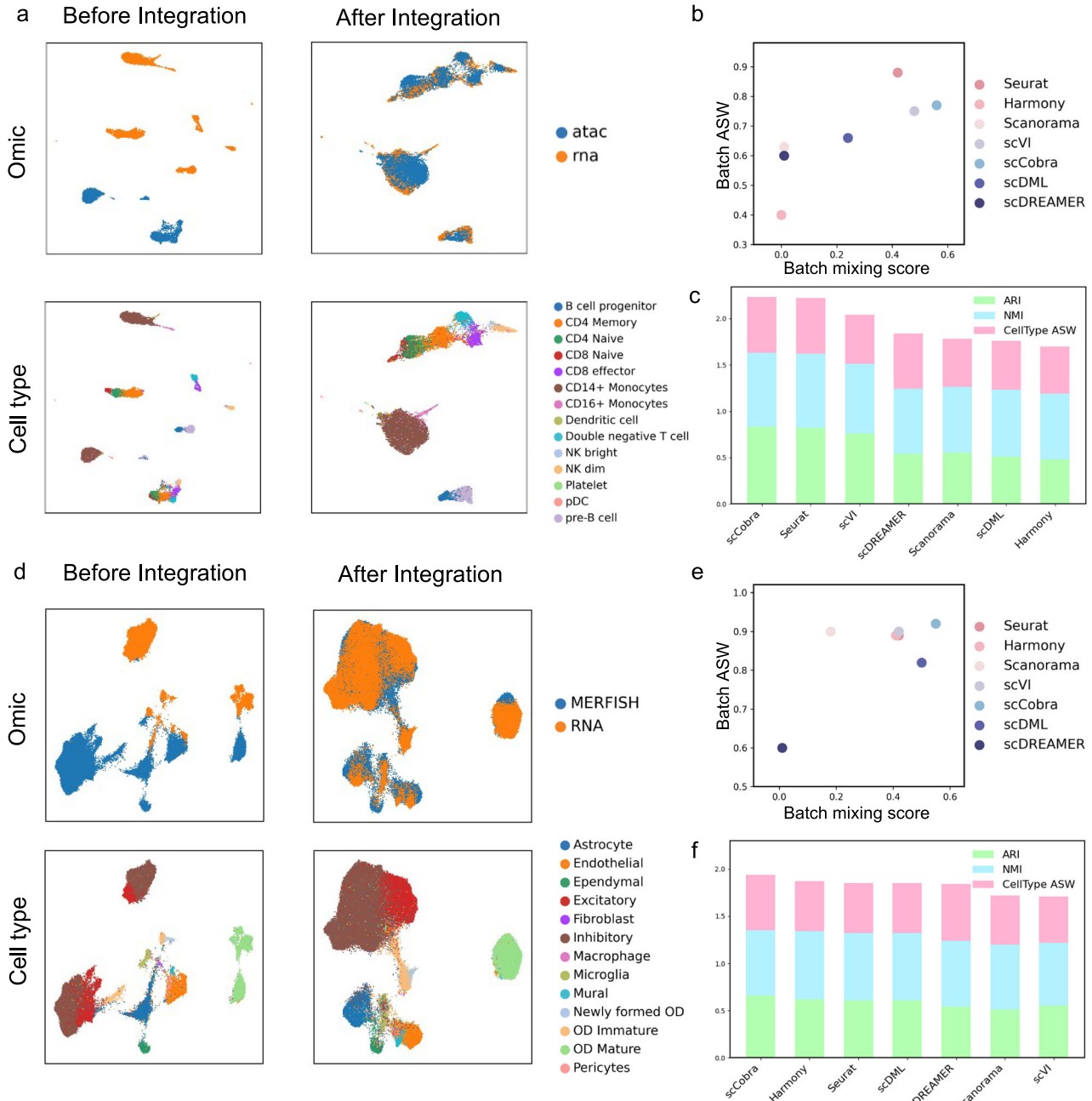

**Fig. 4 | Multimodal batch correction. a** The UMAP visualization results before and after batch correction of the single-cell multi-omic dataset. In the first row, the colors represent the different omics, while in the second row, the colors represent the cell types. **b** The dot plot illustrates the ability of scCobra and benchmarked methods to integrate multi-omic data. The horizontal axis represents the Batch mixing score, while the vertical axis represents the Batch ASW score. Methods located in the upper right corner demonstrate the ability to integrate batches and effectively remove batch effects. **c** The bar plot illustrates the performance of scCobra and benchmarked methods in retaining biological information when integrating multi-omic data, measured by the sum of three biological conservation metrics: ARI, NMI, and CellType ASW. Higher sums indicate better retention of biological information. **d** The UMAP visualization results of the single-cell spatial dataset before and after batch correction. In the first row, the colors represent the MERFISH data and RNA-seq data respectively, while in the second row, the colors represent the cell types. **e, f** The dot plot and bar plot show the performance of scCobra compared to competing methods in integrating spatial omic data, similar to the evaluation methods in (**b, c**).

from the simulation dataset and used scCobra to generate data for one of the batches. The generated data was seamlessly integrated with other benchmarking datasets (refer to Supplementary Fig. S10).

**scCobra provides a flexible online label-transfer framework**

Label transfer is a crucial data harmonization task and often requires consideration of batch effects between unlabeled data and reference datasets. Therefore, many conventional methods[13,14,16] first perform batch correction on the reference atlas and unlabeled dataset and then employ classification methods such as KNN to transfer the labels from the reference dataset to the unlabeled dataset. Consequently, retraining is necessary each time a new dataset needs to be annotated, resulting in repetitive consumption of computational resources. In the case of annotating multiple new datasets, the computational burden is further increased. To address this limitation, scCobra provides an online integration strategy that enables seamless integration without the need for retraining on unlabeled datasets.

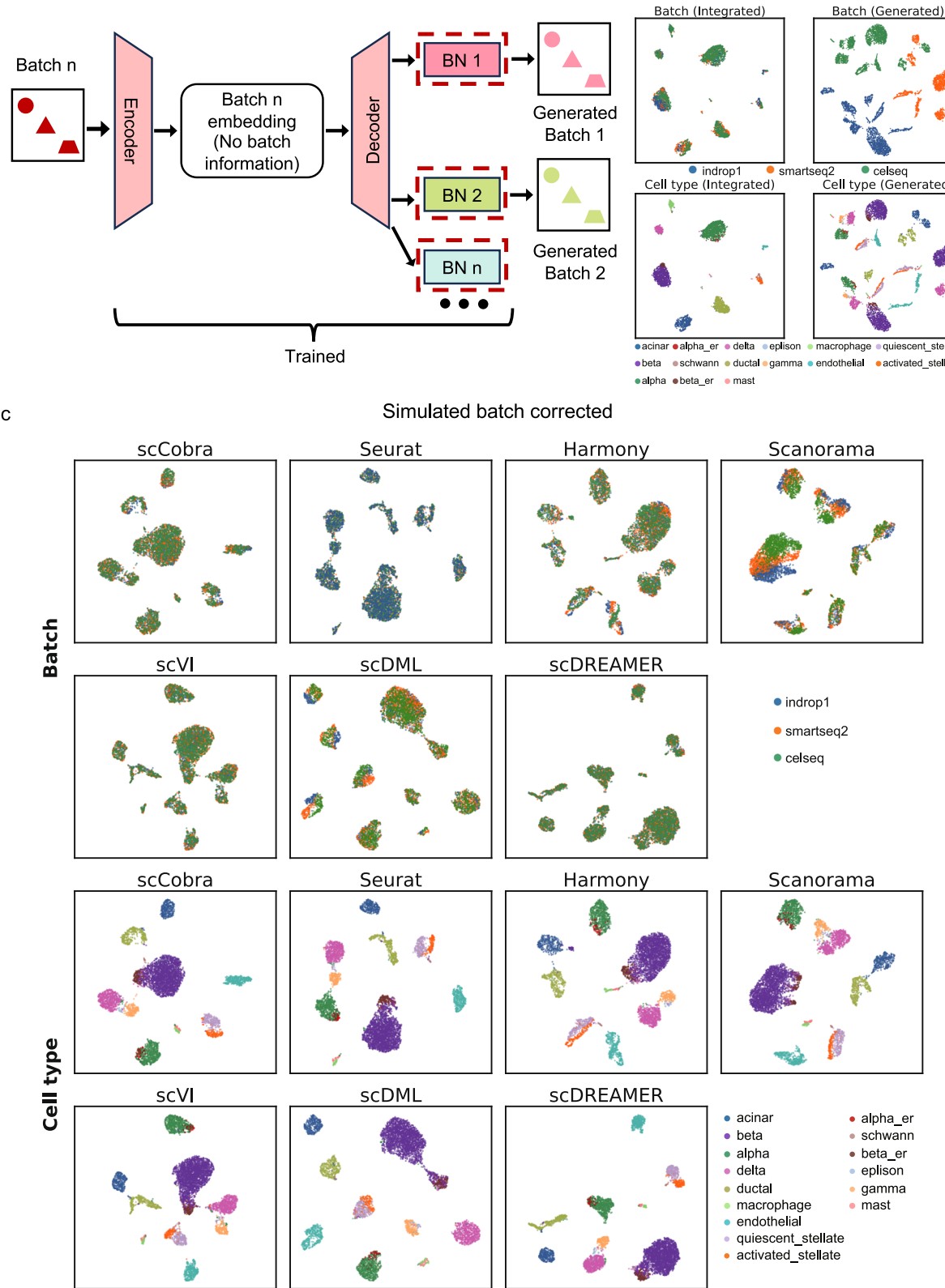

**Fig. 5 | scCobra's efficacy in simulating scRNA-seq data with batch effects.**
**a** Illustration of scCobra's capacity to generate scRNA-seq data simulations that include batch effects, showcasing the adaptability and application of the scCobra model. **b** Through UMAP visualization, this section demonstrates the effectiveness of scCobra in removing batch effects from data, as well as showcasing real scRNA-seq data re-generated with batch labels, highlighting its practical utility in real-world scenarios. **c** Features UMAP visualizations of the simulated data after batch correction illustrate the integration of batches in the top two rows, while the bottom two rows demonstrate the clustering of cells of the same type, with these panels designed to illustrate the efficiency of various methods in correcting batch effects within datasets simulated by scCobra. This simulation acts as a benchmarking to evaluate the performance of other batch correction tools such as Seurat, Harmony, Scanorama, scVI, scDML, and scDREAMER, as shown in (**c**).

Specifically, scCobra first integrates multiple batches of scRNA-seq datasets to construct a reference atlas and saves the trained model. The pre-trained model is then used to directly infer unlabeled scRNA-seq data, mapping them onto the constructed reference atlas. Annotation is performed using a KNN classifier trained on the labels of the reference atlas. This workflow allows scCobra to annotate new datasets without the need to retrain the model. To demonstrate its effectiveness, we first performed batch integration on the human pancreas scRNA-seq dataset used in the batch correction benchmarking and used the corrected results to construct a reference atlas. The integrated data were saved, and a KNN classifier was trained on the labels of the reference atlas. Leveraging the pre-trained model, we then inferred the new datasets GSE114297, GSE8547, and GSE8339, mapping them onto the constructed reference atlas (Fig. 6a). Subsequently, the KNN classifier was applied to annotate these new datasets, seamlessly integrating them with the reference.

In addition to the comparison with SCALEX[26], we also utilized the VAE-based method scVI and transformer-based method TOSCIA to annotate the query dataset. scVI effectively disentangles batch information and true expression values in the raw expression data. We performed joint training of the raw data from the pancreas reference dataset and query dataset, facilitating the transfer of labels from the pancreas reference dataset to the query datasets. TOSICA[58] employs gene pathways as a mediator to directly learn the relationship between gene expression levels and corresponding cell types. We employed a pre-trained TOSICA model, trained on the pancreas reference dataset, to directly infer cell type labels for the query dataset. Notably, both scCobra and SCALEX achieved a weighted F1 score of 0.95 (Fig. 6b, c). In contrast, scVI achieved a weighted F1 score of 0.92 (Fig. 6d), while TOSICA attained a weighted F1 score of 0.94 (Fig. 6e). These findings underscore the exceptional online annotation capability of scCobra.

## The contrastive learning head, discriminator head, and adaBN are all indispensable for the remarkable batch correction performance

We conducted ablation experiments to dissect the key components of scCobra, examining how the removal of specific modules—contrastive learning, adversarial learning, and adaptive Batch Normalization (adaBN)—impacts its overall performance. These modified versions of scCobra were tested for batch effect correction on both Pancreas scRNA-seq and Brain scATAC-seq datasets. UMAP visualizations revealed that removing either the contrastive or adversarial learning modules significantly reduced the efficacy in correcting batch effects within the Pancreas scRNA-seq dataset. Conversely, omitting the adaBN module yielded batch correction visualizations on par with the complete scCobra model (Fig. 7a). On the Brain scATAC-seq dataset, models without contrastive or adversarial learning were unable to effectively correct batch effects. Interestingly, the model without adaBN not only differentiated between Cerebellar Granule Cells and Excitatory Neurons but also surpassed the full scCobra model's performance in doing so (Fig. 7b). Further, we assessed the batch correction effectiveness of these four configurations using quantitative metrics. On the Pancreas scRNA-seq dataset, the scCobra model consistently excelled across various metrics (Fig. 7c). Yet, in analyzing the Brain scATAC-seq dataset, while the complete scCobra setup outperformed others in Batch ASW and Batch_mix_score, highlighting its superior batch correction capacity, it slightly lagged behind the adaBN-excluded model in terms of NMI and ARI scores (Fig. 7d). This suggests that, in terms of preserving biological characteristics, the contribution of adaBN is not substantial.

To examine the impact of adaBN from different angles, we carried out further ablation experiments, this time focusing on aligning single-cell datasets from varied origins. These experiments revealed that without adaBN, the model struggled to merge multi-omic data, as indicated by the failure to blend RNA and ATAC signals. On the other hand, the complete scCobra model, incorporating adaBN, successfully addressed batch effects in multi-omic datasets (Fig. 7e). Additionally, adaBN emerged as critical for effective label transfer in scCobra, enhancing its utility. While the model

without adaBN could cluster cells of the same type, it was less effective in accurately relocating cells from the GSE81547 and GSE114297 batches back to their original reference datasets (Fig. 7f). These findings highlight adaBN's vital role in dynamically aligning data from multiple sources into a cohesive space, an ability that proves essential as differences across datasets become more pronounced.

## Discussion

In this study, we developed a deep neural network-based scRNA-seq batch correction and data integration method that employs contrastive learning and GAN, which outperforms other state-of-the-art methods in scRNA-seq data batch correction in terms of multiple batch correction evaluation metrics (Supplementary Table 1). We applied scCobra in different scRNA-seq datasets that represent distinct scRNA-seq batch correction tasks, and the scCobra batch-corrected scRNA-seq data preserves the biological information while the batch effects are minimized (as documented by the biological conservation and batch correction metrics that we benchmarked).

scCobra utilizes a deep neural network framework that integrates contrastive learning into the VAE–GAN architecture. This is designed to address the common challenges in scRNA-seq data integration and harmonization, particularly those arising from batch effects. Here we outline the key contributions of scCobra below. (1) Over-correction risk often impedes the effective adaptation of batch correction methods and can weaken the biological signal after correction, making it challenging to discern true biological variations from artifacts. scCobra exhibits a significantly reduced risk of over-correction compared to MNN-based methods, which is crucial for preserving the integrity of biological signals. This lower risk enhances scCobra's ability to accurately distinguish between different disease states and identify authentic differential genes, ensuring that vital biological insights are maintained and not obscured during the batch correction process. (2) scCobra addresses complex batch effects in multimodal data, a challenge that many batch correction methods face. Unlike addressing batch effects in multi-batch scRNA-seq, the disparities within multi-omic data are considerably more pronounced. We applied a modality conversion strategy to transform scATAC-seq data into a gene activity matrix and reframed the issue of multimodal data integration as a multi-batch integration problem. This approach has potential benefits, by converting data from other modalities into scRNA-seq, it means that we can utilize existing mature scRNA-seq analysis pipelines[13,59] to handle the data in a unified manner and conduct downstream analyses[60–62]. On the other hand, methods such as scGlue[31] that directly integrate multimodal data in the latent space are relatively more challenging when it comes to understanding multimodal data at the gene level. Demonstrations of scCobra's capability show it adeptly managing batch effects in tasks that integrate scRNA-seq with scATAC-seq, and in those combining scRNA-seq with spatial data, underscoring its robustness and versatility in handling diverse data types. (3) scCobra's online label-transfer functionality enables the integration and annotation of new data batches without the necessity for model retraining. This feature contrasts with many models that require joint training with a reference dataset, a process that can consume substantial computational resources. By bypassing the need for retraining, scCobra streamlines the data harmonization process essential for various downstream tasks, thus facilitating the efficient analysis of large-scale scRNA-seq datasets. (4) Unlike other deep neural network-based methods such as scVI[16] and scANVI[63] which operate under specific prior assumptions regarding the gene expression distribution (e.g., negative binomial or zero-inflated negative binomial), scCobra, employing contrastive learning within its framework, does not necessitate any preconceived notions about gene expression distribution. This flexibility is particularly advantageous, addressing the challenge of finding a universally applicable gene expression distribution for diverse application scenarios. (5) Unlike Splatter[64], which simulates batch effects through multiple sampling, scCobra uses a DSBN layer to learn batch information from real data and adds the learned batch information to scRNA-seq data from the same batch to generate data with real batch effects. This means that scCobra understands and models the

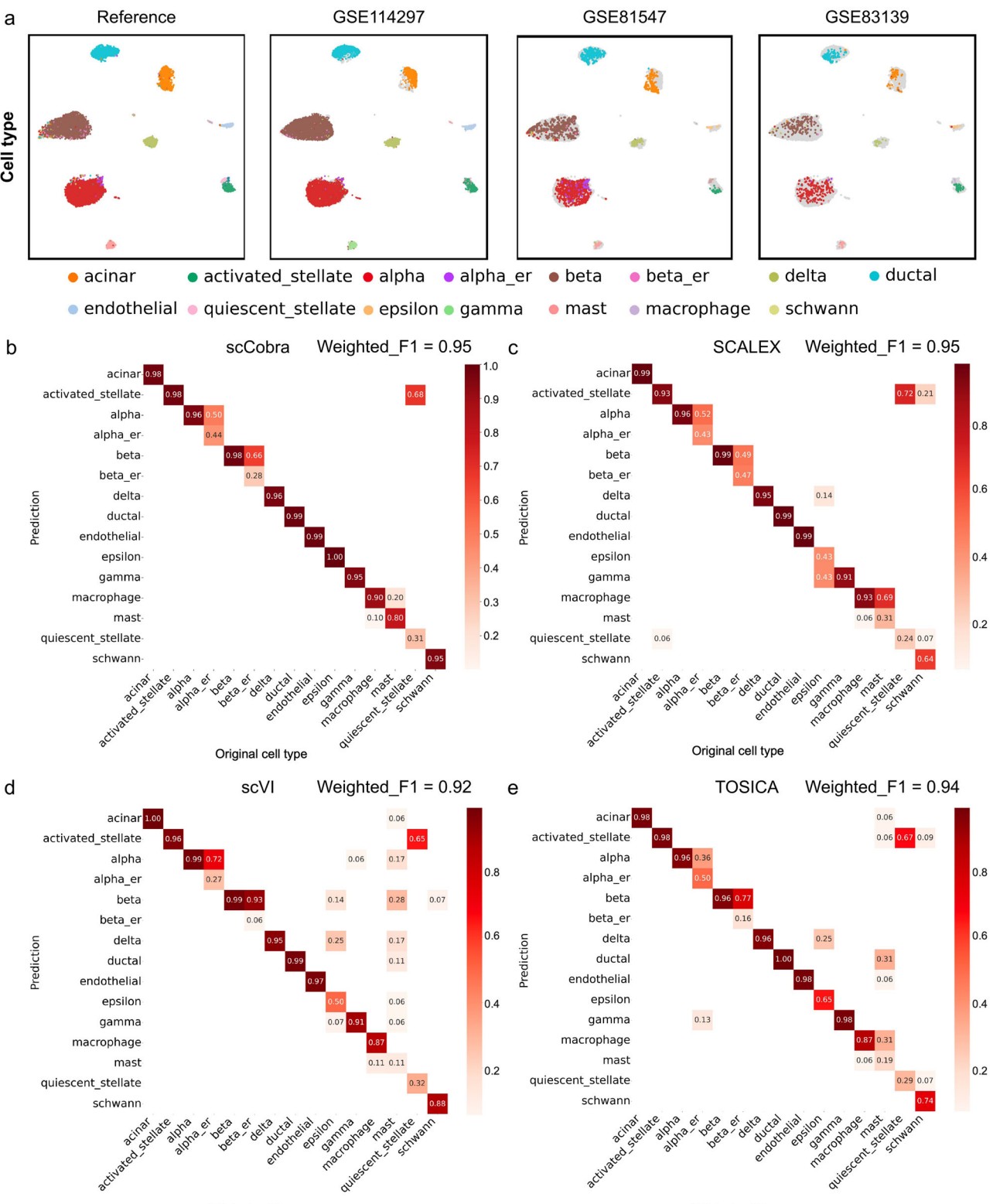

**Fig. 6 | Label transfer for data harmonization across three different studies using scCobra. a** Features UMAP visualization results: the left side shows the reference atlas constructed using scCobra, while the right side displays integrated data from three different studies (GSE114297, GSE85447, and GSE83139), mapped onto the reference atlas and colored by cell type. This illustrates scCobra's capability in aligning datasets from distinct studies through batch effect removal for consistent label transfer. The label-transfer results for scCobra (**b**), SCALEX (**c**), scVI (**d**), and

TOSICA (**e**), comparing the effectiveness of each method in transferring labels from the reference atlas to the unlabeled datasets derived from the three studies. The color intensity in each panel reflects the annotation accuracy (ACC), with darker colors indicating higher ACC, the numbers indicate the specific values of ACC for cell-type annotations. This comparison highlights the utility of scCobra in facilitating online label transfer and data harmonization across various scRNA-seq datasets from different studies.

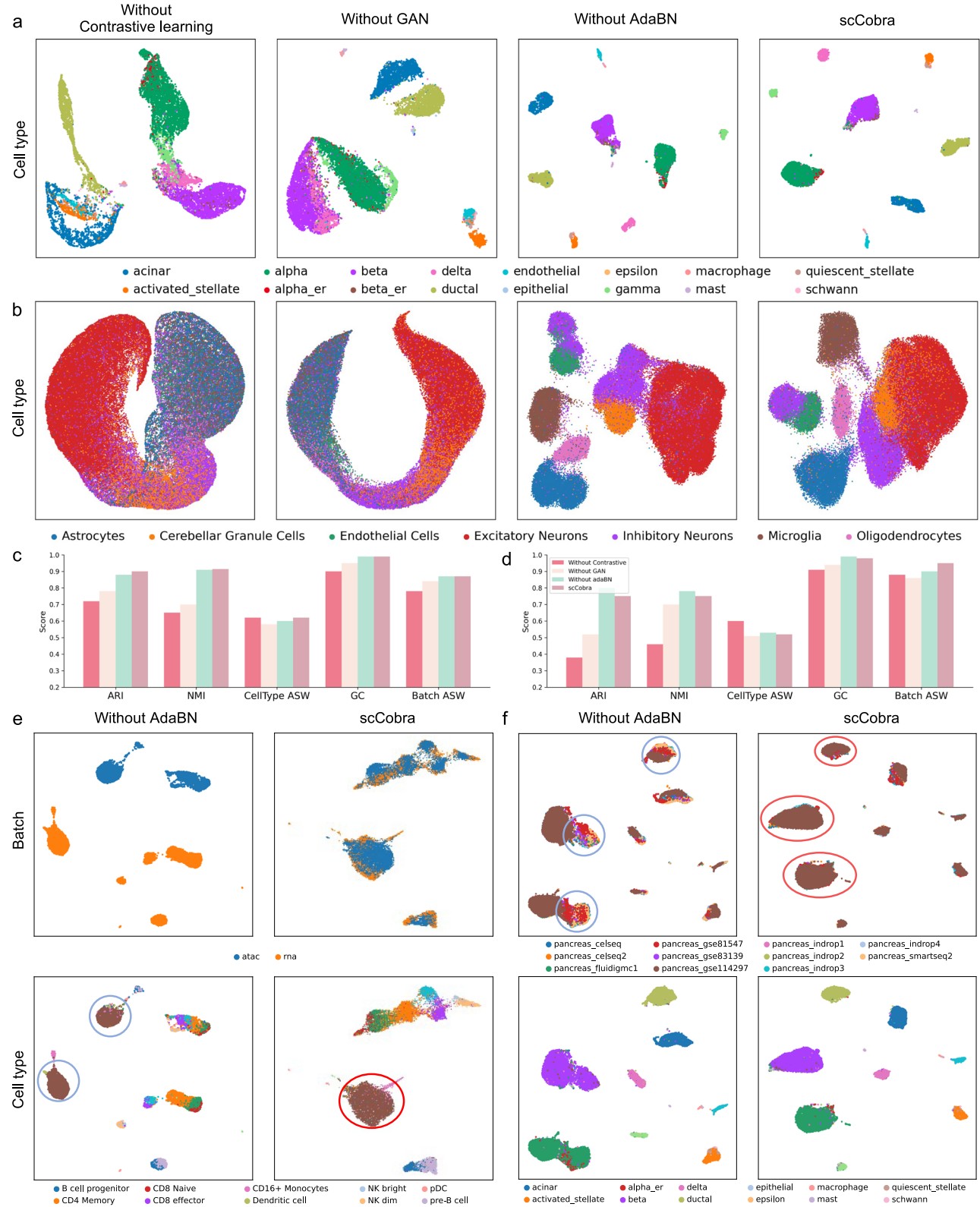

**Fig. 7 | Ablation results for all components.** UMAP visualization results of model correction pancreas scRNA-seq dataset (**a**) and brain scATAC-seq dataset (**b**) excluding contrastive learning, excluding GAN, excluding adaBN and complete scCobra, colored by cell type. Quantification results of three ablation models and scCobra integrated pancreas scRNA-seq dataset (**c**) and brain scATAC-seq dataset (**d**), colored by model. **e** The results of integrating multi-omic data by the model without adaBN, colored by Batch and CellType respectively. Models without AdaBN

failed to integrate *CD14+* Monocytes, which are highlighted with a blue circle. In the scCobra model, *CD14+* Monocytes have been integrated and are marked with a red circle. **f** The result of label transfer for a model that does not include adaBN, colored by Batch and CellType respectively. In the UMAP plot depicting batch effects, models without AdaBN were unable to uniformly integrate data from multiple batches, as indicated by the blue circle, whereas the scCobra model achieved a more uniform integration of batches, which is highlighted with a red circle.

batch effects in real data, making it more suitable for generating test datasets to evaluate the batch-removing capabilities of benchmarked methods.

While scCobra presents notable advantages, it is not without limitations. One such limitation is that scCobra's approach to multimodal integration relies on converting scATAC-seq peak matrices into gene activity matrices, which is contingent upon the algorithm used for this conversion process and may potentially lead to information loss; strategies capable of performing omics transformation, such as scCross[65], BABEL[66], and scButterfly[67], may help address this issue. Additionally, scCobra relies on a shared encoder for integration, which requires selecting shared features between modalities. However, common single-cell protein data, such as CITE-seq, typically contain only a few hundred features, making it challenging to identify shared features and optimize the model effectively. Furthermore, the label transfer performed by scCobra requires that the reference and query datasets originate from the same tissue type or contain similar cell types to ensure optimal performance. Constructing the reference atlas with data from multiple diverse batches further enhances scCobra's generalization ability, enabling more robust and accurate label transfer. Addressing these limitations could be a focus for future work to further improve scCobra's performance and versatility.

scCobra is tailored to tackle multiple challenges in integrating and harmonizing single-cell data, especially those arising from batch effects. Its ability to integrate heterogeneous single-cell data enhances the comprehension of intricate biological mechanisms and varied cellular conditions, thereby reducing obstacles to integrating, analyzing, and leveraging large-scale scRNA-seq datasets.

## Methods

### Data preprocessing

In preprocessing our scRNA-seq data, we first filtered out cells that expressed fewer than 200 genes and genes that were detected in fewer than 3 cells. Additionally, we removed cells where mitochondrial genes accounted for more than 5% of the total gene expression. The normalization of this filtered dataset was conducted using Scanpy's normalize function, which standardized library sizes across the samples. This was followed by a log1p transformation to stabilize the variance. To further refine our analysis and focus on the most informative features, we identified 2000 highly variable genes for downstream analysis. Additionally, scCobra normalized the input data to ensure that the model's input values were scaled within the range of 0–1.

### The scCobra pipeline

scCobra employs multi-phase training to facilitate the convergence of neural networks and more effectively direct gradients to specific layers of the network[68]. During the training process, the scCobra pipeline integrates the VAE–GAN architecture, contrastive learning, and DSBN unfolding, with optimization performed in three distinct phases.

**Phase 1**. The domain discriminator ($Dis^d$) is a batch label discriminator that determines whether the output of the encoder contains batch information. Its optimization goal is defined as:

$$L_d = \min_{\omega} \sum_{i=1}^{N} CE\left(\underset{\omega}{Dis}^d\left(E(\mathbf{x}_i)\right), \mathbf{b}_i\right) \tag{1}$$

Where $E$ represents the encoder, $Dis^d$ represents the batch label discriminator, $\mathbf{x}_i$ represents a single cell, $\mathbf{b}_i$ represents the batch label, $CE$ represents the cross-entropy loss, $N$ is the number of cells in the single-cell dataset.

The generative discriminator is similar to the conventional GAN discriminator, which can differentiate between generated and real data. Therefore, the optimization goal defined as follows:

$$L_D = \max_{\varphi} \sum_{i=1}^{N} \left(\log\left(\underset{\varphi}{Dis}^g\left(E_f(\mathbf{x}_i)\right)\right) + \log\left(1 - \underset{\varphi}{Dis}^g\left(E_f(D(E(\mathbf{x}_i)))\right)\right)\right) \tag{2}$$

Where $Dis^g$ represents the generative discriminator, $E$ represents the encoder, $D$ represents the decoder, $E_f$ represent the fixed encoder (its parameters are consistent with the encoder $E$), $\mathbf{x}_i$ represents a single cell.

**Phase 2**. After the optimization of the discriminator, we can train the generator in a GAN-like manner. Specifically, we maximize the previously trained discriminators. Since data reconstruction in the VAE–GAN framework is generated through the encoder and decoder, this step also optimizes the encoder and decoder. The optimization objective is defined as follows:

$$L_G = \min_{\rho, \gamma} \left(\sum_{i=1}^{N} \log\left(Dis^g\left(E_f\left(\underset{\gamma}{D}\left(\underset{\rho}{E}(\mathbf{x}_i)\right)\right)\right)\right) - \sum_{i=1}^{N} CE\left(Dis^d\left(\underset{\rho}{E}(\mathbf{x}_i)\right), \mathbf{b}_i\right)\right) \tag{3}$$

Where $Dis^g$ represents the previously trained generative discriminator from Phase 1. $E$ represents the encoder, $D$ represents the decoder, $Dis^d$ represents the batch label discriminator, $\mathbf{x}_i$ represents a single cell, $\mathbf{b}_i$ represents the batch label, $CE$ represents the cross-entropy loss.

**Phase 3**. In order to maintain consistency between the original input and reconstructed output, cell-level alignment and cluster-level alignment are necessary. To achieve this, we optimize using contrastive learning loss.

To find a good contrastive learning space, we use the VAE encoder ($E_f$, fixed parameters) to map both the original data and the reconstructed data into a latent space, then the output of $Cell_i$ (original cell's expression, $\mathbf{x}_i$) and $Cell_i'$ (reconstructed cell's expression, $\mathbf{x}_i'$) would be:

$$\mathbf{h}_i = E_f(\mathbf{x}_i), \quad \mathbf{h}_i' = E_f(\mathbf{x}_i') \tag{4}$$

Then the loss of cell-level contrastive head and cluster-level contrastive head would be:

$$l_{cell} = -\log \frac{\exp(s(Dis^c(\mathbf{h}_i), Dis^c(\mathbf{h}_i'))/\tau_c)}{\sum_{j=1}^{N}\left[\exp(s(Dis^c(\mathbf{h}_i), Dis^c(\mathbf{h}_i'))/\tau_c) + \exp\left(s\left(Dis^c(\mathbf{h}_i), Dis^c\left(\mathbf{h}_j'\right)\right)/\tau_c\right)\right]} \tag{5}$$

$$l_{cluster} = -\log \frac{\exp(s(Dis^l(\mathbf{h}_i), Dis^l(\mathbf{h}_i'))/\tau_l)}{\sum_{j=1}^{M}\left[\exp(s(Dis^l(\mathbf{h}_i), Dis^l(\mathbf{h}_i'))/\tau_l) + \exp\left(s\left(Dis^l(\mathbf{h}_i), Dis^l\left(\mathbf{h}_j'\right)\right)/\tau_l\right)\right]} \tag{6}$$

$$L_c = \frac{1}{N}\sum_{i=1}^{N} l_{cell} + \lambda \frac{1}{M}\sum_{i=1}^{M} l_{cluster} \tag{7}$$

$$s(\mathbf{a}, \mathbf{b}) = \frac{\mathbf{a} \cdot \mathbf{b}^T}{\|\mathbf{a}\|\|\mathbf{b}\|} \tag{8}$$

Where $l_{cell}$ represents the cell-level contrastive loss of the original sample, $l_{cluster}$ represents the cluster-level contrastive loss, $M$ is the number of expected clusters in the single-cell dataset, $L_c$ represents the sum loss of contrastive learning, $s(\mathbf{a}, \mathbf{b})$ represents the cosine similarity between two vectors, $Dis^c$ and $Dis^l$ represents the cell-level contrastive head and cluster-level contrastive head respectively, $\tau_c$ and $\tau_l$ represents the temperature coefficient of cell-level contrastive head and cluster-level contrastive head, respectively, $\lambda$ represents a weight coefficient.

In addition, we also need to add KL loss and reconstruction loss (Binary Cross Entropy loss) in VAE for constraints, which is defined as follows:

$$L_{VAE} = \frac{1}{N}\left(\sum_{i=1}^{N}\left[-\mathbf{x}_i \cdot \log(\mathbf{x}_i') - (1 - \mathbf{x}_i) \cdot \log(1 - \mathbf{x}_i')\right] + KL\left(q(\mathbf{z}_i \mid \mathbf{x}_i)\|\mathcal{N}(0, \mathbf{I})\right)\right) \tag{9}$$

Where $\mathbf{x}_i$ represents the original cell's expression and $\mathbf{x}_i'$ represents the reconstructed cell's expression. The first part of the formula corresponds to the reconstruction loss, while the second part represents the KL Divergence Loss. Here, $q(\mathbf{z}_i|\mathbf{x}_i)$ represents the latent distribution of the data, and $N(0,\mathbf{I})$ denotes the prior distribution, which is a standard normal distribution.

The total loss for Phase 3 is defined as:

$$L_{sum} = L_c + \lambda_{vae} \cdot L_{VAE} \qquad (10)$$

After completing the three-step optimization process described above, a full optimization cycle is achieved, and scCobra completes the optimization after multiple rounds of iteration.

The DSBN module in scCobra, located after the decoder, is designed to restore the batch-specific information from the original data. To achieve this, DSBN records the mean and variance of the original data for each batch. The mean ($\boldsymbol{\mu}_B$) and variance ($\boldsymbol{\sigma}_B^2$) of each batch are computed directly from the original data ($\mathbf{x}_i$) within that batch. For a batch containing $B$ cells, the mean and variance are calculated as follows:

$$\boldsymbol{\mu}_B = \frac{1}{B}\sum_{i=1}^{B}\mathbf{x}_i, \quad \boldsymbol{\sigma}_B^2 = \frac{1}{B}\sum_{i=1}^{B}(\mathbf{x}_i - \boldsymbol{\mu}_B)^2 \qquad (11)$$

Here, $\mathbf{x}_i$ represents the original data corresponding to each sample in the batch. This approach ensures that batch-specific statistics are accurately captured, enabling effective batch restoration in the downstream process.

The model architecture parameters of scCobra are detailed in Supplementary Table 2, while the temperature coefficients and weight parameters are provided in Supplementary Table 3. Further details on DSBN are provided in the Supplementary Materials.

## Cell-level contrastive learning and cluster-level contrastive learning

scCobra simultaneously employs cell-level contrastive learning and cluster-level contrastive learning, the key difference between them lies in their granularity and optimization focus, as clearly illustrated in Fig. 1. Contrastive clustering[69], which inspired our approach, effectively combines instance-level and cluster-level contrastive learning to generate meaningful image representations while simultaneously enabling clustering. Building on this concept, we integrated both levels of contrastive learning into the single-cell VAE–GAN framework, resulting in the development of scCobra.

**Cell-level contrastive learning.** At the single-cell level, the focus is on aligning the representations of the same cell under different views (e.g., input and reconstruction). In Fig. 1, these are represented as positive pairs (connected by "pull" arrows), while all other cells in the minibatch are treated as negative samples (pushed away, as shown by "push" arrows). The goal is to ensure that the latent representations of the same cell are close while separating them from representations of other cells.

**Cluster-level contrastive learning.** At the cluster level, the focus shifts to aligning representations of clusters. The cluster-level contrastive head aggregates the outputs of cells to form cluster representations. Positive pairs are representations of the same cluster, while negative pairs are representations from different clusters, as depicted in Fig. 1. The "pull" and "push" arrows indicate the optimization objective of pulling representations within the same cluster closer and pushing representations from different clusters apart. The cluster-level contrastive head includes a SoftMax classifier that outputs the probabilities of each cell $\mathbf{x}_i$ and $\mathbf{x}_i'$ belonging to $M$ clusters. This results in a $2K \times M$ matrix, where $K$ is the number of cells in the minibatch, and $2K$ accounts for both the original input and the reconstructed output.

To facilitate cluster-level contrastive learning, we first transpose this $2K \times M$ matrix to obtain an $M \times 2K$ matrix. From this, we extract two separate $M \times K$ matrices:

1. One represents the cluster-level embeddings derived from the original inputs.
2. The other represents the cluster-level embeddings derived from the reconstructed outputs.

Using these two $M \times K$ matrices, we can perform contrastive learning at the cluster level. Positive pairs are formed between corresponding clusters (e.g., Cluster $j$ from the original input and Cluster $j$ from the reconstructed output), while clusters from different groups are treated as negatives. This approach enables us to align cluster representations effectively, ensuring consistent structure across the latent space.

In summary, cell-level contrastive learning emphasizes fine-grained alignment of single-cell representations, while cluster-level contrastive learning promotes global structure by distinguishing between clusters. Together, these two levels of contrastive learning ensure that scCobra captures both detailed single-cell characteristics and meaningful cluster-level relationships.

## Batch correction benchmarking

All raw data were subjected to the same preprocessing pipeline, including quality control, normalization, and the selection of 2000 highly variable genes as input features. To ensure fairness in cross-method comparisons, the same filtering criteria were applied for HVG selection across all batch correction and integration approaches, including scCobra, Seurat, Harmony, Scanorama, scVI, scDML, and scDREAMER. For all methods, UMAP computation was performed on the latent embeddings derived from the respective integration models. This standardized approach guarantees that each method is evaluated under uniform conditions.

**Clustering method.** Clustering was conducted using the scIB package[27], which iteratively evaluates clustering performance. The scIB pipeline applied Leiden clustering across a range of resolutions from 0.1 to 2.0 (in increments of 0.1) and selected the clustering result with the highest NMI for downstream analyses.

**Evaluation metrics.** In batch correction evaluation, we use the ARI[70], which measures the agreement between clustering results and known labels, and the NMI[70], which evaluates the similarity between the predicted and true cluster assignments. Additionally, we employ the average silhouette width across cell types (CellType ASW)[71], which assesses the separation and cohesion of clusters to measure the biological conservation of the data. The batch correction performance is further evaluated using the average silhouette width across batches (Batch ASW)[71], which indicates the degree of mixing between batches, and Graph Connectivity, which measures the connectivity of cells within the latent space. All metrics were calculated in the latent space, with higher values indicating better performance. To ensure the robustness of our results, we performed five experiments using different random seeds and generated error bars to quantify the uncertainty of the results.

**Differential gene expression analysis.** We conducted a differential gene expression analysis using two healthy samples (HCC03N and HCC04N) from the liver cancer dataset. Both the original (raw) gene expression data and the batch-corrected data were used for this analysis. Employing the Omicverse toolkit, we focused specifically on T/NK cells to identify DEGs between the two samples. Genes exhibiting a log2FC greater than 0.6 and an adjusted $P$ below 0.05 were considered significantly differentially expressed. For this experiment, we used Seurat, Harmony, Scanorama, and scVI as benchmarking methods with default parameters.

## Over-correction evaluation

**Simulated perturbation.** We utilized the immune dataset to simulate a scenario for examining the risk of over-correction. This process involved retrieving genes associated with the "Defense Response to Virus" GOBP

term from NCBI. After normalizing the dataset and applying a log1p transformation, we focused on 2000 highly variable genes. Subsequently, we isolated *CD4*+ T cells from batch "10X" and increased the expression value of each gene associated with the "Defense Response to Virus" GOBP term by 1 within this subset. This process generated a set of perturbed *CD4*+ T cells, distinct from the original cells, allowing us to assess the potential impact of over-correction in the context of enhanced viral defense gene expression. In addition, we performed a similar simulation on *CD14*+ monocytes from batch "10X," perturbing the expression of genes derived from the "Reactome Influenza Life Cycle" Reactome pathway. For this experiment, we used Seurat, Harmony, Scanorama, scVI, scDML, and scDREAMER as benchmarking methods with default parameters to evaluate their performance in mitigating over-correction.

**Normalized over-correction score**. In the evaluation of over-correction, we employed the normalized over-correction score to quantify the severity of over-correction. The over-correction score[26] measures the percentage of inconsistencies in cell types within the neighborhood of each cell, with higher scores indicating a more severe mixing of cell types and, consequently, a greater extent of over-correction. To calculate the normalized over-correction score, we first computed the over-correction score for the raw data as a baseline. Then, we subtracted this baseline score from the scores of all batch correction methods. This normalization ensures that the metric reflects over-correction relative to the noise inherent in the raw data, enabling a more accurate comparison across methods. To enhance the reliability of our results, we repeated the calculation of the over-correction score using five different random seeds and averaged the outcomes to account for variability. The scores were derived from two-dimensional UMAP embeddings, following the methodology described in the original SCALEX[26] paper.

**Survival analysis**. We first obtained gene expression data from the GSE149614 dataset[52], focusing specifically on cancerous tissue samples (HCC01T) and normal tissue samples (HCC03N). To account for potential batch effects, we applied several batch correction methods, including Seurat, Harmony, Scanorama, scVI, and scCobra, to the original expression matrix. After batch correction, we performed differential gene expression analysis on both the raw and corrected datasets using the Omicverse toolkit[72]. In this analysis, the disease (cancerous) samples served as the experimental group, while the healthy (normal) samples served as the control group. For each batch correction method, we identified DEGs and then selected the top 75, top 50, and top 25 genes based on their statistical significance. Next, we performed a survival analysis following the Scissor approach[73]. Using bulk data from TCGA, this method evaluates the significance of DEGs in distinguishing high-risk and low-risk patient groups. Specifically, patients were stratified into high-risk and low-risk groups based on the expression levels of the top-ranked DEGs. Finally, we calculated survival-adjusted *P* to evaluate the statistical significance of the survival differences between these two groups. For this experiment, we used Seurat, Harmony, Scanorama, and scVI as benchmarking methods with default parameters.

**GOBP enrichment analysis**. In the GOBP analysis of the COVID-19 dataset, we focused on identifying differences between healthy and disease samples. For the raw data, we first combined three healthy samples (AP1, AP2, and AP3) with three disease samples (BGCV01_CV0902, BGCV01_CV0904, and BGCV02_CV0902) and performed log normalization on the combined dataset, followed by the selection of highly variable genes. *CD4*+ T cells were then extracted from all samples for differential gene expression analysis. Specifically, *CD4*+ T cells from the healthy samples were treated as the control group, while those from the disease samples were treated as the treatment group. Differential expression analysis was conducted using Omicverse[72] to identify significantly upregulated genes, which were subsequently used for GOBP

enrichment analysis to highlight the top 10 enriched pathways. It is important to note that, since all batch correction methods had already preprocessed the original data, no additional preprocessing was applied to the corrected gene expression data before performing the differential analysis. For this experiment, we used Seurat, Harmony, Scanorama, and scVI as benchmarking methods with default parameters.

**Multimodal batch correction**
In the multi-omic data preprocessing stage, we converted the open chromatin profiles from scATAC-seq into a gene activity matrix utilizing the MAESTRO package. We treated the gene activity matrix and the original scRNA-seq dataset as two independent batches for the purpose of batch correction. Cells expressing fewer than 200 genes and genes found in less than 3 cells were filtered out. We normalized the total counts per cell to 10,000 and applied a log1p transformation to alleviate the effects of data sparsity. To improve the signal-to-noise ratio in these datasets, we selected 2000 highly variable genes for further computational analysis. After completing data preprocessing, we performed multimodal batch correction in the same manner as the integration of multiple batches of scRNA-seq data. The evaluation compared the performance of multi-omic batch correction methods, including Seurat, Harmony, Scanorama, scVI, scDML, and scDREAMER.

In the evaluation of multi-omic integration, we use the ARI[70], NMI[70], and average silhouette width across cell types (CellType ASW)[71] to measure the biological conservation of the data. The batch correction performance is assessed using the average silhouette width across batches (Batch ASW)[71] and the Batch Entropy Mixing Score (Batch mixing score)[26]. The Batch Entropy Mixing Score is a quantitative metric designed to evaluate the homogeneity of cell distribution across different batches in single-cell transcriptomic data. It accounts for batch size discrepancies and calculates local batch mixing by assessing the entropy of nearest-neighbor distributions across multiple iterations. Higher values of this score indicate better batch integration. All metrics were computed in the latent space, with higher values reflecting superior performance.

**Generation of datasets with controllable batch effects**
To generate scRNA-seq data with real batch information, scCobra first trains the model by integrating scRNA-seq data from multiple batches. The preprocessing of the data follows the same approach used during batch correction to ensure consistency in data handling. The trained encoder is capable of removing batch effects, while the decoder, equipped with the DSBN layer, can restore batch information, each BN (Batch Normalization) layer within the DSBN layer is used to restore the batch information for a specific batch. At this point, by selecting the source of the batch information for the generated data, the specific BN layer for that batch in Decoder can be activated. Consequently, the decoder can generate scRNA-seq data with the specific batch information. The evaluation metrics and calculation methods used in benchmarking with scCobra-generated scRNA-seq data are consistent with those applied in the batch correction benchmarking process.

**Online label transfer**
Online label transfer involves two stages: first, scCobra integrates a multi-batch dataset to construct a reference atlas and trains a KNN classifier using the true cell labels from the reference dataset. Second, the query dataset, which is independent of the reference atlas, is processed using the pre-trained encoder to remove batch effects in the latent space, mapping the query dataset onto the constructed reference atlas. The pre-trained KNN classifier is then used to annotate the query dataset, ensuring the model generalizes effectively to unseen data. We computed the label-transfer accuracy (ACC) for each cell type individually, normalizing it based on the number of cell types, and filtered out misclassified points with proportions below 0.05. A heatmap was generated to visually represent these results. Given the varying cell type proportions, we used the weighted F1 score[70] to evaluate the overall accuracy of the label annotations, where the weight for each cell type corresponds to its relative abundance. Finally, we compared scCobra's label-transfer performance with SCALEX, scVI, and TOSICA.

## Implementation of benchmarking methods

In our benchmarking analysis, we focused on evaluating a series of computational tools for their effectiveness in the integration and analysis of scRNA-seq data. Our evaluation highlighted their capabilities in addressing batch effects, facilitating data integration, and improving interpretability. (1) Seurat (v3.2.3): employed for data normalization and the selection of 2000 highly variable genes using the "vst" method, allowing for precise identification of significant features across batches. (2) scVI (scvi-tools, v0.11): utilized for its advanced batch correction techniques within a VAE model by adjusting the "n_batch" parameter, demonstrating its prowess in minimizing batch discrepancies. (3) Harmony (Harmony-pytorch v1.0) and Scanpy: these tools were combined for cell and gene filtering, with Harmony applied after PCA to harmonize datasets, ensuring seamless integration. (4) Scanorama (v1.6): integrated with Scanpy for merging datasets, it employed the "Seurat" method for selecting highly variable genes, creating an integrated analysis matrix essential for downstream analysis. (5) SCALEX (v1.0): chosen for its ability to integrate multiple batches of datasets through label-transfer functions post-training, SCALEX is crucial for maintaining biological coherence. (6) TOSICA (v1.0.0): utilized for its direct annotation capabilities using a reference atlas, enabling accurate identification of cell types in unlabeled data batches, thus enhancing the interpretability of the results. (7) scDML (v0.0.1): by default, 15 clusters were selected, and batch correction was performed using the "rule2" parameter to ensure consistent data integration. (8) scDREAMER: employed for batch correction using all default parameters. Through this exercise, we aimed to provide a comprehensive evaluation of each tool's performance, shedding light on their utility and effectiveness in scRNA-seq data integration and harmonization.

## Statistics and reproducibility

All statistical data are presented as mean ± SD unless otherwise specified. A $P < 0.05$ was considered statistically significant. Each experiment was conducted at least three times unless stated otherwise. All analyses were performed using Python 3.10.

## Reporting summary

Further information on research design is available in the Nature Portfolio Reporting Summary linked to this article.

## Data availability

In this study, we used three real scRNA-seq datasets to evaluate our method, similar to the datasets that were used in the scIB's method[27]. The human lung atlas dataset[27,36], which includes 16 batches, 17 cell types, and over 32,000 cells. The pancreas dataset was composed of 14 cell types mixed across nine sequencing platforms, where each sample of InDrop-seq data was regarded as an independent batch, including 16,382 cells with 19,093 genes, this dataset was collected from Gene Expression Omnibus (GEO) (GSE81076[45], GSE85241[44], GSE86469[44], GSE84133[46], GSE81608[46]). The third immune atlas dataset contains 10 batches with 16 cell types, including 33506 cells with 15148 genes. This dataset is available with the GEO accession number GSE115189[41], GSE128066[54], GSE94820[39], and https://support.10xgenomics.com/single-cell-gene-expression/datasets/3.0.0/pbmc_10k_v3. The simulation dataset was generated by the Splatter package[64]. The dataset was composed of 12,097 cells with 9979 genes, a mixture of seven cell types from six batches. In the over-correction evaluation, beyond the Immune dataset, we also used Liver cancer data from GSE149614[52] and COVID-19 data from https://covid19cellatlas.org/[58]. Multi-omic data are from https://support.10xgenomics.com/single-cell-gene- expression/datasets/3.0.0/pbmc_10k_v3 and https://support.10xgenomics.com/single-cell-atac/datasets/1.0.1/atac_v1_pbmc_10k. The MERFISH dataset has collected data on 64,373 cells with 155 genes, and the scRNA-seq dataset includes 30,370 cells with 21,030 genes[57]. The datasets used for label-transfer evaluation come from GSE83139[74], GSE114297[75], and GSE81547[76]. Numerical source data for all graphs in the manuscript can be found in the Supplementary Data file.

## Code availability

The implementation of scCobra is available at https://github.com/mcgilldinglab/scCobra.

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

## Acknowledgements
This work was partially supported by CIHR PJT-180505, FRQS 295298, 295299, and NSERC RGPIN-2022-04399 to J.D. Y.X. is supported by grants from the National Science Foundation of China (62172274).

## Author contributions
Bowen Zhao, Jun Ding, and Yi Xiong jointly conceptualized the methodology and experimental design of the article. Bowen Zhao, Kailu Song, and Jun Ding collaboratively wrote the manuscript. Dongqing Wei provided support with computational support.

## Competing interests
The author declares no competing interests.
