## [Transparent Peer Review file · Communications Biology]

scCobra allows contrastive cell embedding learning with domain adaptation for single cell data integration and harmonization

Corresponding Author: Dr Jun Ding

Version 0:

Reviewer comments:

Reviewer #1

(Remarks to the Author)

This manuscript proposes a novel batch effect correction method, scCobra, which utilizes contrastive cell embedding learning. The authors compared scCobra with several popular batch effect correction methods, including Seurat, Harmony, Scanorama, and scVI. However, based on the presented results, I am not convinced that scCobra demonstrates superior performance in batch effect removal or clustering accuracy compared to these methods. Below are some of my concerns:

1. In Fig. 2a (over correction evaluation), the perturbed CD4+ cells overlap with the 10X dataset. Was this simulation performed only on the CD4+ cells from the 10X dataset?
2. In Fig. 2d, only scCobra has mixed CD4+ and CD8+ T cells together. These two types of T cell have significant biological differences, which raises the concern that scCobra may be overcorrecting.
3. The explanation of Fig. 2e is unclear. After clustering, Seurat and Harmony should use raw counts to perform differential expression analysis (DE). Why do the raw counts results indicate "defense to virus" while Seurat and Harmony do not capture this?
4. From the results shown in Fig. 6, scCobra does not appear to outperform other methods. For instance, in the simulated and pancreas datasets shown in Fig. 6, scCobra's ARI and NMI scores are lower than those of several other methods. Additionally, why are the UMAP for the immune dataset not included? Given that scCobra performs better than other methods on this dataset, it would be important to show these results.
5. Batch effect removal or integration is one of the most critical steps in single-cell RNA sequencing data analysis, and many excellent methods have been developed in recent years, such as scDREAMER, scDML, and FIRM. I suggest the authors compare scCobra with these methods. Although the manuscript includes comparisons with several methods, the most recent method compared was published in 2019.
6. The authors' description of the evaluation steps is overly brief, lacking crucial details. For instance, how were the UMAP dimensionality reduction results for scCobra obtained? Were they derived from batch removed expression or from the latent space? What clustering method was used, and how was the clustering performed? Did all methods use the same 2000 highly variable genes?

Reviewer #2

(Remarks to the Author)

Zhao et al. introduced scCobra, a deep neural network algorithm combining contrastive learning and domain adaptation for correcting batch effects in single-cell analysis across various platforms and modalities. scCobra excels in batch effect correction while minimizing the risk of over-correction, which is crucial for downstream analyses such as online label transfer and data integration. The authors have benchmarked scCobra against competing methods, demonstrating its superior performance. While the manuscript is well-organized overall, there are several points that require clarification or improvement.

Major comments:

1. Since scCobra is primarily designed for batch effect correction, it might be better by prioritizing the batch effect correction results in Figure 6 earlier in the result section, to emphasize the main goal of scCobra.

2. Please provide more details on how the quantitative metrics were calculated for each method, particularly for scCobra. Were these metrics computed based on the latent representations of VAE, or from the two-dimensional embeddings by UMAP? If the latter, it's important to note that UMAP may not fully preserve data structures in two-dimensional space, potentially compromising the reliability of quantitative metrics as indicators of model performance.
3. In Figure 2a, cells of the same type appear in different layers across datasets (e.g., CD14 Monocytes). This raises questions about whether the shift between CD4+ T cells and perturbed CD4+ T cells is due to cell state changes or batch effects. Consider including more perturbed cell types to better demonstrate scCobra's effectiveness in addressing over-correction issues.
4. The GOBP analysis in the COVID-19 dataset requires further explanation: a) Please provide more details on how the GOBP analysis was performed on raw data. b) Explain why raw data is used as a ground truth, given potential batch effects in raw data. c) Clarify why the "Defense response to Virus" pathway was spotted on here in COVID-19 data, when it was verified in the perturbed human immune dataset but not in COVID-19 data. d) Why the shared terms found by other methods (e.g., "Neutrophil activation involved in immune response" by scVI and Seurat, "cytokine-mediated signaling pathway" by Harmony) could not be used to explain the batch effect correction performance? It's not convincing to use raw data as ground truth to validate scCobra's superiority in batch effect correction. Instead, employing a use case with known ground truth, such as simulated perturbed 10X data, would provide a more reliable benchmark.
5. The UMAP visualizations in Figure 4c compare the batch correction performance of various methods. Given that the simulated data is well-clustered by cell type, simulating single-cell data with continuous underlying structures might better demonstrate scCobra's strengths in batch effect correction.
6. In the online label transfer section, if the model is jointly trained on both reference and query datasets (as stated by "We performed joint training of the raw data from the pancreas reference dataset and query dataset, facilitating the transfer of labels from the pancreas reference dataset to the query datasets."), the metrics may reflect training accuracy rather than generalization ability. To better assess generalization, it would be helpful to train the model only on reference data and test annotation accuracy on the query datasets.

Minor comments:

1. As scCobra is a deep learning-based model, a more detailed description of its architecture is necessary. Please clarify the number of hidden layers in the encoder, decoder, and discriminators, as well as the number of neurons in each layer, the dimension of latent factors, the temperature coefficient in contrastive learning loss, and the trade-off parameters lambda used in the loss functions.
2. The datasets used in the results should be clarified clearly. If they're scRNA-seq data, please use "scRNA-seq data" instead of "single-cell datasets".
3. Explain how the over-correction score is computed and normalized. What do negative values in the results signify?
4. In Figure 2b, the expression values of IR27 and TLR9 seem to only contain two values, which is uncommon for scRNA-seq data. Please verify the expression data for these genes and explain why they were chosen for demonstration. Are these genes perturbed in the data, or are they differentially expressed across CD4+ T cell states?
5. The positioning of OD immature and OD mature cells in Figure 3d does not appear as close as suggested by the manuscript. Please review the figure and corresponding statements for consistency.
6. Provide more details on the simulated datasets generated by Splatter. In Figure 4b, although six batches are simulated, only two batches are used in Figure 3c. Consider omitting unnecessary batches from the figures. Additionally, clarify the goal of batches 3-6 as they appear to be clustered by cell type without batch effects.
7. The figure descriptions should focus on describing the contents rather than interpreting the results. For example, in Figure 4b, explain what "normalized over-correction score" means. Is a higher score better? Why are batches 1 and 2 separated from batches 3-6 in the plots? What's the metric used in figure 5b-e to measure accuracy?
8. Annotate data source in figure 3a and figure 3d for better understanding.
9. In the loss function of domain discriminator and GAN, should x_i and b_j use the same index i ?
10. What's the difference between contrastive learning at the single-cell level and the cluster level? Both l_{cell} and $l_{cluster}$ use x_i and x_i' at single-cell level.
11. Figure 7a, it seems like the cells are colored by cell type not batch. If so please correct titles.
12. Correct typos in manuscript, e.g., "minimizes" rather "mininizes", "modality" rather "modal".

Reviewer #3

(Remarks to the Author)

In the field of single cell data analysis, many methods have been developed to correct batch effects. However, these methods are limited by the prior assumptions of gene distribution and have the problem of overcorrection. This study proposes a new method for batch correction, based on the neural network architecture of VAE-GAN and the strategy of contrastive learning. This method can not only solve the problem of batch overcorrection, but also realize multimodal batch correction, simulating the batch effects, and label transfer.

Overall, this is an interesting study and this new approach could be a useful tool and a great complementary to the field. The work is technically strong and the manuscript is well-written and presented. I have a few questions to be addressed that are listed below.

Major points:

Can the authors generate simulated datasets with more scenarios to further evaluate the method? For example, a dataset contains at least two batches with imbalanced cell subpopulation compositions (i.e., one cell group is specific to one of the batches), and a dataset has high batch complexity with nested sub-batches (i.e., one batch may contain three sub-batches). In the model architecture diagram, there are two encoders. The manuscript mentions that the parameters of encoder E_f are

fixed and consistent with the parameters of encoder E. In the optimization process mentioned in the manuscript, the first step is to optimize the two discriminators, and the updated parameters are the parameters of the discriminator. The second step is to optimize the generator, at which time the parameters of the encoder and decoder are updated. I am wondering whether the parameters of encoder E and E_f are shared here? In addition, the contrast loss, reconstruction loss and KL loss are introduced in the third step. The gradient return of these losses will also affect the parameters of the encoder and decoder. Are the second and third step carried out successively? In general, VAE uses KL loss and reconstruction loss to return gradients when optimizing and updating parameters. Why does the manuscript optimize the second and third steps separately? Finally, is the reconstruction loss the MSE? It will be better if the authors can write the mathematical formula in more detail in the method.

In contrast loss, h_i in $[\dots]$ _cluster should be the representation of a cluster. How can we get the representation of the entire cluster from the representation of a cell in a cluster?

Can you give a more specific formula interpretation for the BN block designed for the model? For example, how is the mean and variance of each batch determined? Is it to record the mean and variance of the original data or the mean and variance of the reconstructed data?

Minor points:

The work only considers RNA and ATAC in multimodal batch integration. Is it applicable to other modalities such as RNA and Protein?

For the label transfer task, the authors mention that there is no need to retrain on new data, only pre-training on a reference data. Is there any requirement for the quality of the reference data, such as data volume, labels, etc.?

Some text on the figures are too small, particularly the figure legends.

Version 1:

Reviewer comments:

Reviewer #1

(Remarks to the Author)

Thank the author for the careful revision of the manuscript and for thoughtfully addressing each of my questions. I have no further questions and recommend the article for publication.

Reviewer #2

(Remarks to the Author)

The manuscript has improved substantially following the authors' comprehensive revisions. However, there remain a few points that require further clarification:

1. Please enlarge the figure legends and text size in both the main figures and supplementary figures to enhance readability.
2. It is unclear whether a higher or lower value of the overcorrection score is considered optimal in supplementary fig.6. Please specify any criteria or thresholds used to identify the best results.
3. The first plot in Figure 3b is labeled "IR27" instead of "IL27." Kindly double-check and correct any similar errors throughout the manuscript.

Reviewer #3

(Remarks to the Author)

The authors have well addressed all my comments. I have no further comments. Great work!

We would like to express our gratitude to the reviewers and editorial team for the constructive comments and suggestions offered on our manuscript, titled "scCobra: Contrastive cell embedding learning with domain-adaptation for single-cell data integration and harmonization" with manuscript ID COMMSBIO-24-4123-T. The detailed feedback from the reviewers and the editorial team has substantially improved the manuscript's quality and thoroughness. We also thank the editor for their expedient and careful handling of our submission. Following the guidance provided, we have addressed each comment in detail. Enclosed are our point-by-point responses to the concerns raised and the amendments applied to the manuscript (all changes were tracked in blue).

Reviewer #1

General Comments: This manuscript proposes a novel batch effect correction method, scCobra, which utilizes contrastive cell embedding learning. The authors compared scCobra with several popular batch effect correction methods, including Seurat, Harmony, Scanorama, and scVI. However, based on the presented results, I am not convinced that scCobra demonstrates superior performance in batch effect removal or clustering accuracy compared to these methods. Below are some of my concerns:

Response: Thank you for your thoughtful feedback on our manuscript. We appreciate your comments on the performance benchmarking and unique contributions of scCobra. In response, we have made significant updates to the manuscript to clarify the advantages of scCobra and address your concerns. Below, we highlight key contributions of scCobra from three comprehensive perspectives:

1. Enhanced Benchmarking to Demonstrate scCobra's Superiority in batch correction across key metrics

To strengthen our benchmarking analysis, we have incorporated additional methods (refer to **Reviewer #1 C5** response), expanded the datasets (see response to **Reviewer #1 C4**) to include biologically relevant real-world examples such as the human lung atlas¹, and conducted evaluations using multiple random seeds to ensure result robustness. Our updated results demonstrate that scCobra achieves superior performance across key metrics, including Adjusted Rand Index (ARI), Normalized Mutual Information (NMI), CellType ASW, and Graph Connectivity (GC), while maintaining Batch ASW scores comparable to existing methods (see **Supplementary Table 1**). These results underscore scCobra's ability to correct batch effects effectively while preserving critical biological signals and cellular heterogeneity.

We further tested scCobra's performance in another scenario by comparing the number of differentially expressed genes (DEGs) between samples under the same condition, where no significant biological differences are expected. Using raw data from two healthy samples in the liver cancer dataset, we identified 470 DEGs, reflecting substantial batch noise. After batch correction, scCobra again demonstrated superior performance by reducing the number of DEGs to

30, compared to scVI (66) and Harmony (40). These results further highlight scCobra's robustness in effectively removing batch noise more efficiently than other methods, while preserving meaningful biological signals (See response to Reviewer #1 C3).

2. Minimizing Over-Correction Risks

Over-correction, which leads to the loss of meaningful biological variation, is a significant limitation of many batch correction methods. We demonstrate scCobra's ability to minimize over-correction risk through the following experiments (Supplementary Table 1):

(1) scCobra successfully captures implanted biological signal that was masked by over-correction in other methods (See response to Reviewer #1 C1)

We demonstrated scCobra's ability to preserve biological signals through a simulated perturbation experiment using the Immune dataset. Specifically, we perturbed the expression of a subset of genes in *CD4+* T cells, creating perturbed and unperturbed cellular states. After applying batch correction, we observed that existing methods, including Seurat, Harmony, scDML², and scDREAMER³, failed to effectively distinguish between the perturbed and unperturbed states, indicating a high risk of over-correction. In contrast, scCobra minimized this risk by clearly separating the perturbed and unperturbed *CD4+* T cells, demonstrating its robustness in maintaining biological distinctions.

To ensure these results were not cell type-specific or perturbation-specific, we repeated the experiment on *CD14+* monocytes. Similarly, scCobra successfully differentiated perturbed and unperturbed counterparts, whereas Seurat, Harmony, scVI, scDML, and scDREAMER failed. Additionally, scCobra consistently exhibited lower over-correction scores compared to these methods, further validating its ability to reduce the risk of masking true biological variation during batch correction.

(2) scCobra minimizes over-correction risk, as demonstrated by the survival analysis, in real-world datasets (See response to Reviewer #1 C3 for complete details)

In the liver cancer dataset⁴, survival analysis demonstrated that differentially expressed genes (DEGs) identified from gene expression data corrected by scCobra and other existing methods effectively stratified patients into low-risk and high-risk groups, with scCobra achieving the best performance. Specifically, scCobra's survival-adjusted P was 4.1×10^{-21} , markedly outperforming scVI (1.1×10^{-19}), Seurat (9.6×10^{-11}), Scanorama (4.4×10^{-14}), Harmony (2.4×10^{-9}), and raw data (1.0×10^{-8}). These results highlight scCobra's superior ability to reduce batch effects without compromising the biological signals critical for downstream analyses, thereby mitigating the risk of over-correction.

3. scCobra offers a broad spectrum of functionalities missing from existing methods (refer to the Supplementary Table 1)

In addition to its primary role in batch correction, scCobra also provides a range of unique functionalities that distinguish it from existing methods and make it a versatile tool for single-cell analysis.

First, scCobra includes the ability to generate synthetic datasets with controlled batch effects. This functionality is particularly valuable for benchmarking and validation purposes, as the synthetic datasets generated by scCobra share the same origin and come with robust ground truth labels. These datasets enable comprehensive evaluations of batch correction methods under controlled conditions, facilitating the development and testing of new methodologies (Refer to our manuscript result section "*scCobra Enables Simulation of Single-Cell Data with Batch Effects*" for further details). Second, scCobra supports efficient label transfer without requiring model retraining. By learning shared latent spaces across batches, scCobra can map new datasets onto reference atlases constructed from multi-batch datasets, enabling consistent cell-type annotation and integration. This capability significantly reduces computational overhead and enhances scalability for large-scale single-cell data analyses (Refer to our manuscript result section "*scCobra provides a flexible online label-transfer framework*" for further details). Third, scCobra integrates datasets across different modalities, making it applicable to single-cell multi-omics studies (Refer to our manuscript result section "*scCobra enables multi-omic batch correction*" for further details). This functionality broadens the scope of analyses by allowing users to incorporate diverse types of data, such as single-cell RNA-seq and ATAC-seq, into unified representations that preserve biological complexity. Finally, scCobra incorporates corrections in the original gene expression space, ensuring that biological signals remain interpretable in their native format. This feature is critical for downstream analyses, such as differential expression and pathway enrichment, where preserving the structure and integrity of the original data is essential.

These unique functionalities position scCobra as a comprehensive solution for batch correction and downstream single-cell analysis, providing capabilities that extend beyond those offered by existing methods. By addressing multiple challenges in single-cell analysis, scCobra not only improves batch correction but also facilitates accurate and scalable integration and harmonization of single-cell datasets across diverse experimental conditions.

Supplementary Table 1: benchmarking table

Method	ARI	NMI	CellType ASW	Graph Connection	Batch ASW	Over-correction	Multi-omic integration	Label Transfer	Batch Generation	Original-space Correction
scCobra	1/7	1/7	1/7	1/7	4/7	2/7	1/7	✓	✓	✓
Seurat	6/7	6/7	4/7	4/7	2/7	5/7	1/7	✓	×	✓
Harmony	4/7	5/7	4/7	5/7	7/7	5/7	7/7	×	×	✓
Scanorama	5/7	3/7	2/7	6/7	1/7	1/7*	6/7	×	×	✓
scVI	2/7	2/7	6/7	3/7	4/7	3/7	3/7	✓	×	✓
scDML	7/7	7/7	5/7	7/7	6/7	7/7	6/7	×	×	×
scDREAMER	3/7	4/7	3/7	2/7	2/7	3/7	4/7	×	×	×

*: Scanorama has the lowest over-correction score, but it tends toward under-correction.

✓: Native implementation. ×: No native implementation provided. 1/7: Ranked 1st out of 7 methods.

C1: “In Fig. 2a (over correction evaluation), the perturbed $CD4^+$ cells overlap with the 10X dataset. Was this simulation performed only on the $CD4^+$ cells from the 10X dataset?”

Response: Thank you for your question about this critical detail. Yes, the simulation was performed only on the $CD4^+$ cells from the 10X dataset. To better illustrate the data's origin, we generated a magnified visualization of the perturbed $CD4^+$ T cell data, highlighting the batch and cell type labels (Supplementary Fig. S4).

Supplementary Fig. S4: scCobra is capable of distinguishing perturbed $CD4^+$ T cells from their unperturbed counterparts. a, Cells are colored by batch, demonstrating that perturbed $CD4^+$ T cells originate exclusively from the 10X batch. b, Cells are colored by cell type. This figure shows that perturbed cells are specific to the 10X batch.

The rationale behind this simulation approach is to implant a biologically meaningful signal into the data, which should remain detectable after batch correction. The loss of this implanted signal indicates over-correction, as the implanted signal becomes masked by the batch correction. Specifically, we perturbed genes associated with the "Defense response to virus" GOBP term in $CD4^+$ T cells, generating a perturbed $CD4^+$ T cell population with distinct cellular states compared to the normal, unperturbed $CD4^+$ T cells. After batch correction, these two populations should remain clearly distinguishable, with minimal overlap. Greater overlap between the two populations indicates a higher degree of over-correction. As shown in Fig. 3a, UMAP visualizations generated by various methods after applying batch correction demonstrate a comparison of their ability to distinguish between perturbed and normal $CD4^+$ T cells. While scVI and Scanorama were also able to differentiate between perturbed and normal $CD4^+$ T cells similarly to scCobra, both methods exhibited limitations. Scanorama showed a tendency toward under-correction, with cells of the same type, such as $CD4^+$ T cells, failing to integrate cohesively and instead appearing scattered across the UMAP space. This lack of proper integration extended to other cell types, including $CD8^+$ T cells and $CD14^+$ monocytes, underscoring the limitations of Scanorama in achieving robust and consistent data integration. Similarly, while scVI was able to separate perturbed and normal $CD4^+$ T cells, the separation was not as clear or consistent as that achieved by scCobra. Additionally, scVI displayed limitations in maintaining integration within the same cell type across batches, leading to suboptimal clustering in certain regions of the UMAP space. In contrast, scCobra successfully separated perturbed and normal $CD4^+$ T cells with minimal overlap while maintaining cohesive integration of cells within the same cell type, outperforming both Scanorama and scVI. We further illustrate the separation of scCobra-corrected $CD4^+$ T cells and perturbed $CD4^+$ T cells using marker genes shown in Fig. 3b. The results reveal a clear and well-defined boundary between these two cellular states. This separation demonstrates that

scCobra effectively minimizes over-correction risk by preserving the biologically meaningful differences between the perturbed and normal cells. Furthermore, marker genes such as *TLR9*, *HSP90AA1*, and *IL27*—associated with the "Defense response to virus" GOBP term—served as robust differentiators, highlighting scCobra’s ability to maintain critical biological variation after batch correction.

Fig. 3: scCobra Can Minimize Over-correction Risk. **a**, Batch correction methods (Seurat, Harmony, Scanorama, scVI, scDML, scDREAMER, and scCobra) were assessed on a simulated immune scRNA-seq dataset with batch effects and biological differences introduced by perturbing genes associated with the "Defense response to virus" GOBP term. The UMAP visualization is colored by cell type. The results demonstrate that scCobra effectively distinguishes perturbed *CD4+* T cells from their unperturbed counterparts during batch integration, with significantly lower levels of over-correction compared to other methods. **b**, Enlarged UMAP plots clearly show that scCobra effectively separates normal *CD4+* T cells from perturbed *CD4+* T cells. This distinction is highlighted by specific markers such as *TLR9*, *HSP90AA1*, and *IL27*, demonstrating that differential markers can reliably distinguish perturbed cells from normal cells. This figure underscores scCobra's ability to maintain critical biological variation after batch correction.

In addition to the above visualizations, we further systematically quantified the extent of over-correction for each method using the over-correction score⁵ (Fig. 3c) to demonstrate the superiority of scCobra in minimizing over-correction risk. When evaluating the dataset containing all cell types, scDML exhibited the highest degree of over-correction, followed by Seurat, Harmony, and scDREAMER, which demonstrated varying levels of over-correction. Among the methods evaluated, scCobra showed a relatively low over-correction score (0.043), outperforming scVI (0.086) and significantly improving over methods like scDML. Although Scanorama reported the lowest over-correction score (0.012), its UMAP visualization (Fig. 3a) revealed a tendency toward under-correction, as it failed to cohesively integrate cells of the same cell type. Focusing only on the subset of *CD4+* T cells and perturbed *CD4+* T cells, we also independently calculated over-correction scores. Seurat, Harmony, and scDREAMER demonstrated over-correction scores exceeding 0.35, while scVI (0.158) and scDML (0.278) also exhibited higher scores compared to scCobra, which achieved a notably lower score of 0.038. Although Scanorama again reported the lowest score (0.0), its under-correction tendency, as seen in the UMAP visualization (Fig. 3a), compromised the biological integrity of the data.

Fig. 3 scCobra Can Minimize Over-correction Risk. **c**, Quantitative over-correction analysis on all cell types and specifically for *CD4+* T cells and perturbed *CD4+* T cells. Higher values indicate a greater degree of over-correction. The results demonstrate that scCobra significantly reduces the risk of over-correction compared to existing methods, as indicated by its lower over-correction scores.

In summary, the separation of *CD4+* T cells and their perturbed counterparts in the UMAP visualization, along with the quantitative over-correction score, highlights scCobra's ability to effectively minimize the risk of over-correction while achieving accurate batch correction and preserving biologically meaningful variation, striking an optimal balance.

To ensure that our conclusions regarding over-correction risk are not confined to a specific cell type and perturbation (i.e., a specific GO term that we have chosen), we extended our perturbed simulation to include another cell type (i.e., *CD14+* monocytes) and perturbation (e.g., "Reactome influenza life cycle"). Specifically, we perturbed genes associated with another pathway "Reactome influenza life cycle" in *CD14+* monocytes from the Immune dataset's 10X batch (Supplementary Fig. S5). The batch and cell type labels of the perturbed *CD14+* monocytes were highlighted to facilitate a detailed evaluation of scCobra and other methods in mitigating over-correction.

Supplementary Fig. S5: UMAP visualization of *CD14+* Monocytes and Perturbed *CD14+* Monocytes from the simulated immune dataset. **a**, Cells are colored by batch, demonstrating that perturbed *CD14+* Monocytes originate exclusively from the 10X batch. **b**, Cells are colored by cell type. This figure shows that perturbed cells are specific to the 10X batch, highlighting their batch-dependent origin.

After performing batch correction, we compared scCobra with Seurat, Harmony, Scanorama, scVI, scDML, and scDREAMER on the perturbed *CD14+* monocyte dataset. The corrected data were then visualized using UMAP, as shown in Supplementary Fig. S6b. The results showed that Seurat, Harmony, scVI, scDML, and scDREAMER failed to adequately separate perturbed *CD14+* monocytes from their normal counterparts, resulting in significant aggregation. In contrast, both

Scanorama and scCobra successfully distinguished the two populations. However, Scanorama exhibited a tendency toward under-correction, as it failed to properly integrate other cell types, such as *CD4+* T cells, *CD8+* T cells, and *CD14+* monocytes, compromising overall data coherence. The UMAP visualization of scCobra-corrected *CD14+* monocytes and their perturbed counterparts revealed a clear separation between the two cellular states (Supplementary Fig. S6c), also highlighting scCobra's ability to minimize over-correction. Marker genes such as *NUP35*, *HSP90AA1*, and *NUP214*, associated with the "Reactome influenza life cycle" pathway, were identified as robust differentiators between these states, further validating scCobra's capacity to preserve biologically meaningful variations.

Supplementary Fig. S6 scCobra is capable of distinguishing perturbed *CD14+* Monocytes from their unperturbed counterparts. **b**, Batch correction methods (Seurat, Harmony, Scanorama, scVI, scDML, scDREAMER, and scCobra) were assessed on a simulated immune scRNA-seq dataset with batch effects and biological differences introduced by perturbing genes associated with the "Reactome influenza life cycle" Reactome pathway. The UMAP visualization is colored by cell type. The results demonstrate that scCobra effectively distinguishes perturbed *CD14+* Monocytes from their unperturbed counterparts during batch integration, with significantly lower levels of over-correction compared to other methods. **c**, Enlarged UMAP plots clearly show that scCobra effectively separates normal *CD14+* Monocytes from perturbed *CD14+* Monocytes. This distinction is highlighted by specific markers such as *NUP35*, *HSP90AA1*, *NUP214* demonstrating that differential markers can reliably distinguish perturbed cells from normal cells. This figure underscores scCobra's ability to maintain critical biological variation after batch correction.

To complement these qualitative findings, we quantified the extent of over-correction for all methods using the normalized over-correction score (Supplementary Fig. S6e). Across all cell types, scDML exhibited the highest degree of over-correction (0.251), followed by Seurat, Harmony, scVI, and scDREAMER, which displayed varying levels of over-correction. In contrast, both Scanorama (0.06) and scCobra (0.073) achieved lower over-correction scores, indicating their relative effectiveness in mitigating over-correction risk. Focusing specifically on *CD14+* monocytes and their perturbed counterparts, Seurat, Harmony, scVI, scDML, and scDREAMER all had over-correction scores exceeding 0.4. By comparison, Scanorama (0.26) and scCobra (0.31)

exhibited significantly lower scores. However, the UMAP visualizations (Supplementary Fig. S6b) revealed that Scanorama’s tendency toward under-correction compromised its ability to adequately separate cellular populations. scCobra, on the other hand, achieved a balance between minimizing over-correction and maintaining robust integration, preserving biologically meaningful distinctions between perturbed and normal cells.

Supplementary Fig. S6 scCobra is capable of distinguishing perturbed CD14+ Monocytes from their unperturbed counterparts. e, Quantitative over-correction analysis on all cell types and specifically for CD14+ Monocytes and perturbed CD14+ Monocytes. Higher values indicate a greater degree of over-correction. The results demonstrate that scCobra significantly reduces the risk of over-correction compared to existing methods, as indicated by its lower over-correction scores.

Overall, our results on CD4+ T cells and CD14+ monocytes collectively validate scCobra's superior ability to minimize over-correction risk while ensuring accurate and biologically coherent integration.

We have updated Fig. 3 and added Supplementary Fig. S4, S5, S6 accordingly. Additionally, we have made the corresponding revisions to the "*scCobra minimizes over-correction risks in batch correction*" section of the manuscript.

C2: “In Fig. 2d, only scCobra has mixed CD4+ and CD8+ T cells together. These two types of T cell have significant biological differences, which raises the concern that scCobra may be overcorrecting.”

Response: Thank you for your insightful observation and the opportunity to clarify this point. The apparent mixing of CD4+ and CD8+ T cells in original Fig.2d (now Supplementary Fig. S8b) was indeed a visual artifact caused by specific plotting parameters, such as point size and transparency settings, rather than an issue with scCobra itself. To address this concern, we reanalyzed the data and adjusted the visualization parameters to enhance clarity. We also performed a separate UMAP visualization of scCobra-corrected CD4+ and CD8+ T cells, as shown in Supplementary Fig. S8, which demonstrates that scCobra does not mix these two biologically distinct populations. Additionally, marker gene analysis further validated this distinction: CD8A (specific to CD8+ T cells shown in Supplementary Fig. S8e), CD4 (specific to CD4+ T cells shown in Supplementary Fig. S8f), exhibited distinct expression patterns in scCobra-corrected data, whereas some overlap was observed in the corrections by Harmony and scVI. The low marker expression values observed for Scanorama and scVI are due to the intrinsic scaling effect of their respective methods, which affects the overall expression levels.

Supplementary Fig. S8: scCobra Can Minimize Over-correction Risk. **d**, Enlarged UMAP plots demonstrate that $CD4+$ and $CD8+$ T cells are clearly separated, with UMAPs colored by cell type. **e**, **f**, Specific markers, such as $CD4$ for $CD4+$ T cells and $CD8A$ for $CD8+$ T cells, effectively identify and distinguish these two cell types. These results show that scCobra preserves the distinct identities of $CD4+$ and $CD8+$ T cells without mixing, avoiding over-correction.

To provide further evidence, we conducted a quantitative analysis using mixing scores⁵, which measure the mixing of different cell types. Higher scores indicate greater difficulty in distinguishing between cell types. scCobra achieved a mixing score between $CD4+$ and $CD8+$ T cells of 0.58, compared to Seurat (0.64), Harmony (0.64), Scanorama (0.53) and scVI (0.59). These results confirm that scCobra offers clearer separation between $CD4+$ and $CD8+$ T cells while minimizing over-correction.

Here we did not include scDML and scDREAMER in the above benchmarking because these methods operate exclusively in the latent space and are designed for batch integration rather than directly correcting raw gene expression data. Since our analysis above required methods capable of batch correction in the original gene expression space, scDML and scDREAMER were not suitable for this evaluation.

We have moved the COVID-19-related results to Supplementary Fig. S8 (refer to response to Reviewer #1 C3). The localized visualization focusing on $CD4+$ and $CD8+$ T cells is now presented in Supplementary Fig. S8. Additionally, we have revised the "*scCobra minimizes over-correction risks in batch correction*" section of the manuscript.

C3: The explanation of Fig. 2e is unclear. After clustering, Seurat and Harmony should use raw counts to perform differential expression analysis (DE). Why do the raw counts results indicate "defense to virus" while Seurat and Harmony do not capture this?

Response: Thank you for this insightful comment. We agree with you that if Seurat and Harmony used the original uncorrected data to identify differentially expressed genes (DEGs), they would have detected pathways consistent with the raw counts results. However, in our analysis, Seurat and Harmony failed to capture the "defense response to virus" signal because we focused on the corrected gene expression in the original space, where the data had already been batch-adjusted.

Consequently, the DEGs identified by these methods differed from those identified using raw counts, resulting in different enriched GOBP terms in original Fig.2e (now Supplementary Fig. S8c).

As you pointed out, given that DEGs derived from raw gene expression can identify the "defense response to virus" term, why not simply use raw gene expression instead of corrected gene expression? The main reason lies in the presence of batch effects in the original data. When identifying DEGs between two conditions (e.g., healthy and diseased samples), batch effects and genuine biological differences are often conflated. This can lead to identifying DEGs that are driven by batch effects rather than true biological signals. While the raw COVID-19 data can indeed detect the "defense response to virus" pathway due to its strong differential signal, this does not mean batch effects are negligible. Batch noise may obscure other critical signals or amplify artifacts, compromising the reliability of downstream analyses. Corrected gene expression mitigates this issue, allowing DEGs to more accurately reflect true biological differences.

To further elucidate the impact of batch noise and highlight the necessity of batch correction, we conducted the following two experiments using the liver cancer dataset:

(1) In the first experiment, we identified differentially expressed genes (DEGs) between samples under different conditions and used these DEGs to perform a survival analysis to distinguish high-risk and low-risk groups. The clearer the separation between the two groups, the more significant and biologically relevant the identified DEGs are, indicating their potential as true disease markers. Conversely, a higher survival-adjusted P suggests that the identified DEGs contain substantial noise originating from batch effects. Specifically, we analyzed the liver cancer dataset⁴ by selecting one normal and one diseased sample for comparison. After applying various batch correction methods, including scVI, Seurat, Scanorama, Harmony, and scCobra, UMAP visualization confirmed that batch effects between the two samples had been effectively removed. This integration validated the technical success of these methods in mitigating batch-induced variability (Fig. 3d). However, the absence of visible batch effects in the UMAP embedding does not necessarily ensure the preservation of meaningful biological signals. To evaluate the impact of batch correction, we conducted survival analysis using sets of differentially expressed genes (DEGs) identified from both raw and batch-corrected data (Fig. 3e), following the Scissor approach⁶. This method, utilizing bulk TCGA data, assesses the significance of DEGs in stratifying patients into high-risk and low-risk groups. Among all approaches, scCobra achieved the strongest performance, evidenced by significantly more robust survival-adjusted P (e.g., adjusted $P = 4.1 \times 10^{-21}$). In contrast, scVI (1.1×10^{-19}), Seurat (9.6×10^{-11}), Scanorama (4.4×10^{-14}), Harmony (2.4×10^{-9}) and raw data (1.0×10^{-8}) produced weaker statistical results, suggesting a loss of crucial biological information. This reinforces the importance of using corrected gene expression to distinguish genuine biological differences from batch noise. To ensure the robustness of our findings, we performed differential gene screening and survival analyses across varying thresholds, yielding consistent results that confirm the reliability of our conclusions (Supplementary Fig. S7).

Fig. 3: scCobra Can Minimize Over-correction Risk. **d**, The UMAP visualizations of the liver cancer dataset, integrated using different methods and colored by cell type, demonstrate the batch correction capabilities of each approach. Compared to the raw data, Seurat, Harmony, Scanorama, scVI, and scCobra all effectively removed batch effects between healthy and diseased samples. **e**, Survival analysis results using differential gene sets obtained from corrected gene expression with scCobra and benchmarking methods. Blue represents high-risk groups, and orange represents low-risk groups, with the x-axis indicating survival days and the y-axis representing the survival proportion. The analysis demonstrates that scCobra-corrected gene expression eliminates batch noise and identifies meaningful differential genes that most effectively stratify patients into high-risk and low-risk groups, improving survival stratification compared to raw and corrected gene expressions from existing methods.

(2) In the second experiment, we compared the number of differentially expressed genes (DEGs) between samples under the same condition. Theoretically, no DEGs should be identified, as no significant biological differences are expected under identical conditions. To illustrate this, we extracted two healthy samples from the liver cancer dataset and found 470 DEGs using the raw data, likely reflecting batch noise rather than true biological variation. However, after applying batch correction, the number of DEGs was significantly reduced: scCobra identified 30 DEGs, scVI identified 66, and Harmony identified 40 (Supplementary Fig. S3). This reduction aligns with the expectation that batch correction effectively removes spurious variations caused by technical artifacts while preserving meaningful biological signals.

Supplementary Fig. S3: scCobra effectively eliminates spurious differential signals caused by batch effects. The figure shows the differences in the number of differentially expressed genes (DEGs) between samples under the same condition, comparing raw gene expression with corrected gene expression using scCobra and other methods. The results demonstrate that scCobra significantly reduces the number of DEGs driven by batch effects, outperforming other methods in eliminating batch-induced noise. These findings confirm that using corrected gene expression data significantly reduces batch noise present in the raw data, thereby restoring true biological signals. As a result, relying on raw counts

for differential expression analysis is not generally recommended, as it may fail to account for the impact of batch noise and could lead to misleading conclusions.

Since the enrichment analysis in our previous manuscript was not systematic enough, we replaced it with more comprehensive survival analysis results. These survival analysis results are now included in the main manuscript (Fig. 3), providing a clearer and more intuitive demonstration of over-correction. This strengthens the evaluation of scCobra's performance against other methods. We have also added Supplementary Fig. S6, S7 to support our conclusions. Additionally, we have revised the "*scCobra minimizes over-correction risks in batch correction*" section of the manuscript to reflect the above changes.

C4: From the results shown in Fig. 6, scCobra does not appear to outperform other methods. For instance, in the simulated and pancreas datasets shown in Fig. 6, scCobra's ARI and NMI scores are lower than those of several other methods. Additionally, why are the UMAP for the immune dataset not included? Given that scCobra performs better than other methods on this dataset, it would be important to show these results.

Response: Thank you for your comment. Due to image size limitations, the UMAP visualization of the Immune dataset is included in the Supplementary Fig. S1a. scCobra effectively aligns multiple batches of the immune dataset while clearly distinguishing various cell types. Quantitative evaluation demonstrates that, compared to existing methods, scCobra preserves the highest level of biological signals (original Fig. 6d, now Fig. 2d).

To address your comments and concern about the performance comparison with other methods, we have undertaken more comprehensive benchmarking efforts. Specifically, we have incorporated additional methods as per suggested in Reviewer #1 C5 response, expanded the range of datasets (We have replaced the simplistic simulated datasets with more biologically relevant real-world datasets, such as the human lung atlas shown in Fig. 2a, while retaining the results of the simulation dataset in Supplementary Fig. S2 for reference.) and for each method, conducted evaluations using multiple random seeds to ensure robustness of the results.

Our expanded quantification results demonstrate that scCobra consistently achieves superior performance across key metrics, including Adjusted Rand Index (ARI), Normalized Mutual Information (NMI), CellType ASW, and Graph Connectivity (GC), while maintaining Batch ASW scores comparable to those of existing methods (refer to the Supplementary Table 1). These findings underscore scCobra's ability to effectively correct batch effects while preserving critical biological signals and cellular heterogeneity.

We further tested scCobra's performance in another scenario by comparing the number of differentially expressed genes (DEGs) between samples under the same condition, where no significant biological differences are expected. Using raw data from two healthy samples in the liver cancer dataset, we identified 470 DEGs, reflecting substantial batch noise. After batch correction, scCobra again demonstrated superior performance by reducing the number of DEGs to 30, compared to scVI (66) and Harmony (40). These results further highlight scCobra's robustness

in effectively removing batch noise more efficiently than other methods, while preserving meaningful biological signals (See response to Reviewer #1 C3).

Supplementary Table 1: benchmarking table

Method	ARI	NMI	CellType ASW	Graph Connection	Batch ASW	Over-correction	Multi-omic integration	Label Transfer	Batch Generation	Original-space Correction
scCobra	1/7	1/7	1/7	1/7	4/7	2/7	1/7	✓	✓	✓
Seurat	6/7	6/7	4/7	4/7	2/7	5/7	1/7	✓	×	✓
Harmony	4/7	5/7	4/7	5/7	7/7	5/7	7/7	×	×	✓
Scanorama	5/7	3/7	2/7	6/7	1/7	1/7*	6/7	×	×	✓
scVI	2/7	2/7	6/7	3/7	4/7	3/7	3/7	✓	×	✓
scDML	7/7	7/7	5/7	7/7	6/7	7/7	6/7	×	×	×
scDREAMER	3/7	4/7	3/7	2/7	2/7	3/7	4/7	×	×	×

*: Scanorama has the lowest over-correction score, but it tends toward under-correction.
 ✓: Native implementation. ×: No native implementation provided. 1/7: Ranked 1st out of 7 methods.

Beyond regular batch correction tasks, the most significant capability of scCobra is its ability to reduce the risk of over-correction during batch integration. We demonstrated this in two sets of experiments.

- (1) The first experiment involved a simulated perturbation study, where we constructed a simulated dataset by perturbing *CD4+* T cells and *CD14+* Monocytes. UMAP visualization showed that scCobra could distinguish between normal and perturbed cells while integrating multi-batch datasets. Quantitative evaluation metrics also supported that scCobra reduced over-correction risks more effectively than other benchmarking methods. (See Reviewer #1 C1 response)
- (2) In the second experiment, we analyzed the liver cancer dataset to further evaluate scCobra's performance. Survival analysis demonstrated that DEGs identified from scCobra-corrected data effectively stratified patients into low-risk and high-risk groups, achieving superior performance compared to other batch correction methods. These results confirm scCobra's robustness in removing batch effects while preserving biologically meaningful signals, thereby enhancing the reliability of downstream analyses. (See Reviewer #1 C3 response).

In summary, the effectiveness of scCobra in batch correction has been rigorously validated across multiple domains, including scRNA-seq and single-cell multi-omics. Furthermore, both simulation studies and analyses of real-world disease datasets have substantiated scCobra's capacity to effectively minimize the risk of over-correction. Beyond its primary role, scCobra also offers a variety of unique functionalities for data integration and harmonization. For example, it supports online label transfer and the generation of synthetic datasets with batch effects, further expanding its utility in downstream analyses.

We have updated the UMAP visualization and benchmarking in Fig. 2 and Supplementary Fig. S1 and added new benchmarking results in Fig. S3. Additionally, we have made revisions to the "*scCobra outperforms benchmarked methods in batch effect correction*" section of the manuscript. We also keep the results of the simple simulation dataset in Supplementary Fig. S2 for reference.

C5: Batch effect removal or integration is one of the most critical steps in single-cell RNA sequencing data analysis, and many excellent methods have been developed in recent years, such as scDREAMER, scDML, and FIRM. I suggest the authors compare scCobra with these methods. Although the manuscript includes comparisons with several methods, the most recent method compared was published in 2019.

Response: Thank you for your suggestions. As per suggested, we have incorporated scDREAMER and scDML into our analysis and re-performed the benchmarking. Due to FIRM's limitation to two-batch integration, we excluded this method, as it does not address the multi-batch scenarios examined in our study.

To strengthen the analysis, we replaced simplistic simulated datasets with complex and biologically relevant real-world datasets, such as the human lung atlas. This new dataset includes 16 batches, 17 cell types, and over 32,000 cells, presenting a challenging scenario for batch correction. The simple simulated datasets were retained in the supplementary material for reference. From the UMAP visualizations (Fig. 2a, b), we observe that scCobra, along with scVI, showed the best performance in separating cell types and integrating batches. In contrast, other methods exhibited notable limitations. For example:

- Harmony mixed Type 2 and Basal 2 cells, and Neutrophil_CD14_high with Macrophages.
- Seurat struggled to separate multiple cell types.
- Scanorama, despite differentiating Basal 1 and Basal 2, failed to integrate Macrophage/Type 2 cells across batches.
- scDML and scDREAMER, while effective in some areas, also showed issues distinguishing specific cell types (e.g., Neutrophil_CD14_high and Macrophages).

Fig. 2 scCobra demonstrates superior batch correction performance over state-of-the-art methods. a, b, UMAP visualizations comparing scCobra with benchmarked methods (Seurat, Harmony, Scanorama, scVI, scDML, and scDREAMER) on the human lung atlas dataset. **Panel a** presents the batch correction results, with different batches distinguished by color, while **panel b** shows cell type aggregation, with each cell type assigned a unique color. Regions of superior performance are highlighted with red circles, indicating areas where scCobra or the benchmarked methods excel, whereas blue circles denote areas requiring improvement.

Quantitatively (Fig. 2c), scCobra achieved the highest ARI, NMI, and CellType ASW scores, confirming its ability to preserve biological signal while integrating batches effectively. In Graph Connectivity scores, scCobra performed comparably to scVI and scDREAMER, while Scanorama achieved the highest Batch ASW score but exhibited under-correction tendencies in UMAP visualizations. Overall, scCobra consistently balanced robust batch correction with the retention of critical biological information. Overall, scCobra demonstrated comparable batch correction performance to other methods while retaining more biological signal, making it the most effective method in this study.

Fig. 2 scCobra demonstrates superior batch correction performance over state-of-the-art methods. c, Quantitative results of scCobra and the benchmarked methods across different datasets. We provide two types of evaluation metrics. Biological conservation metrics include ARI, NMI, and CellType ASW, with higher values indicating better retention of biological signals during batch effect removal. Batch correction metrics include Graph Connection and Batch ASW, where higher values signify better mixing of cells from different batches and more effective removal of batch effects.

Beyond superior batch correction, scCobra’s ability to mitigate over-correction risks represents a key strength, as highlighted in response to Reviewer #1 C1, response to Reviewer #1 C3. This capability has been demonstrated through multiple analyses, including simulated perturbations and survival analysis. Additionally, scCobra extends beyond standard batch correction by offering advanced functionalities such as multi-omic integration, label transfer without retraining, and synthetic dataset generation, making it a versatile and comprehensive tool for single-cell analysis.

In conclusion, we hope we have addressed your concerns by demonstrating scCobra’s robust performance across various metrics and datasets, outperforming both established and recent methods such as scDREAMER and scDML. Its ability to preserve biological signals, minimize over-correction risks, and support diverse applications positions scCobra as a valuable and impactful resource for single-cell genomics research.

We have updated the UMAP visualization and benchmarking in Fig. 2 and Supplementary Fig. S1. Additionally, we have included scDML and scDREAMER as benchmarking methods in the sections "scCobra enables multi-omic batch correction" and "*scCobra enables simulation of scRNA-seq data with batch effects*". Revisions have also been made to all textual descriptions in the manuscript where benchmarking method comparisons are discussed.

C6: The authors' description of the evaluation steps is overly brief, lacking crucial details. For instance, how were the UMAP dimensionality reduction results for scCobra obtained? Were they derived from batch removed expression or from the latent space? What clustering method was used, and how was the clustering performed? Did all methods use the same 2000 highly variable genes?

Response: Thank you for your comment. We have provided additional details to clarify the evaluation steps:

UMAP Computation: scCobra facilitates integration in both latent and original spaces. For benchmarking evaluations, including multi-omic integration, label transfer, and batch generation tasks, UMAP was computed using the latent embeddings produced by scCobra. In contrast, for tasks focusing on over-correction risk, such as survival analysis and enrichment analysis, PCA was applied to the corrected gene expression data in the original space, followed by UMAP computation.

Clustering Method: Clustering was performed using the scIB¹ package, which evaluates clustering results iteratively. scIB applied Leiden clustering across resolutions ranging from 0.1 to 2.0 (in steps of 0.1) and selected the clustering result with the highest Normalized Mutual Information (NMI) for downstream analysis.

Selection of Highly Variable Genes: To ensure consistency and fairness across methods, we applied identical filtering criteria to select 2000 highly variable genes as input. This standardization guarantees that all methods, including scCobra, Seurat, and others, are evaluated under uniform conditions. These steps provide a robust, fair, and reproducible framework for assessing scCobra's performance across various tasks and comparisons.

We have revised the Methods section “*Batch correction benchmarking*” of the manuscript to include detailed descriptions of UMAP computation, Clustering Method, and Selection of Highly Variable Genes.

Reviewer #2

General Comments: Zhao et al. introduced scCobra, a deep neural network algorithm combining contrastive learning and domain adaptation for correcting batch effects in single-cell analysis across various platforms and modalities. scCobra excels in batch effect correction while minimizing the risk of over-correction, which is crucial for downstream analyses such as online label transfer and data integration. The authors have benchmarked scCobra against competing methods, demonstrating its superior performance.

Response: Thank you for your encouraging and thoughtful comments. We are grateful for your recognition of scCobra’s ability to effectively correct batch effects while preserving critical biological signals, a balance that is essential for downstream applications like online label transfer and multi-modal data integration.

In our benchmarking, scCobra consistently outperformed other methods, achieving superior ARI and NMI scores across diverse platforms and modalities. This demonstrates its robustness in integrating datasets while minimizing over-correction risks.

Your feedback reinforces the significance of our approach and motivates us to continue refining scCobra for broader applications in single-cell analysis. We are excited about its potential to

advance the harmonization of datasets across diverse experimental conditions and facilitate new biological discoveries.

C1: Since scCobra is primarily designed for batch effect correction, it might be better by prioritizing the batch effect correction results in Figure 6 earlier in the result section, to emphasize the main goal of scCobra.

Response: Thank you for your constructive suggestion. In response, we have reorganized the manuscript to prioritize the batch correction results in the first part of the Results section, specifically before the benchmarking analyses in Fig. 6. This reorganization allows readers to immediately focus on the core functionality of scCobra in addressing batch effects, which is central to its design and performance. By emphasizing these results early, we aim to strengthen the narrative and better highlight scCobra's impact on single-cell data integration.

We have rearranged the order of the results in our manuscript.

C2: Please provide more details on how the quantitative metrics were calculated for each method, particularly for scCobra. Were these metrics computed based on the latent representations of VAE, or from the two-dimensional embeddings by UMAP? If the latter, it's important to note that UMAP may not fully preserve data structures in two-dimensional space, potentially compromising the reliability of quantitative metrics as indicators of model performance.

Response: Thank you for your insightful comment. To clarify, the quantitative metrics for scCobra and the other benchmarked methods were calculated in the latent space after batch correction. This approach ensures a robust representation of the data that captures underlying biological signals while minimizing batch effects, providing reliable and consistent metric computation. For Seurat, which integrates data in the original space by default, we applied PCA to its output and calculated the metrics in the resulting PCA space. This additional step ensures a standardized evaluation process and a fair comparison across methods.

You are correct that the over-correction score was indeed calculated using the two-dimensional UMAP embeddings. While we acknowledge that UMAP may not fully preserve data structures in two-dimensional space, this strategy, adopted from SCALEX⁵, is widely used in the field and provides a practical and consistent basis for evaluating over-correction. By employing this established methodology, we ensure compatibility with existing practices and facilitate meaningful comparisons with prior results. For more details about the over-correction score, we refer you to Reviewer #2 MC3 and the SCALEX⁵ paper. We hope this explanation resolves your concerns and provides a clear understanding of our evaluation process.

We have revised the descriptions in the Methods section “*Batch correction benchmarking*” and “*Over-correction evaluation*” of the manuscript to specify the calculation methods for each metric, based on either latent representations or 2D-UMAP embeddings.

C3: In Figure 2a, cells of the same type appear in different layers across datasets (e.g., CD14 Monocytes). This raises questions about whether the shift between *CD4+* T cells and perturbed

CD4+ T cells is due to cell state changes or batch effects. Consider including more perturbed cell types to better demonstrate scCobra's effectiveness in addressing over-correction issues.

Response: Thank you for your valuable insights. To ensure that our conclusions regarding over-correction risk are not confined to a specific cell type and perturbation, we extended our perturbed simulation to include another cell type (i.e., *CD14+* monocytes) and perturbation (i.e., "Reactome influenza life cycle"). Specifically, we perturbed genes associated with another pathway "Reactome influenza life cycle" in *CD14+* monocytes from the Immune dataset's 10X batch (Supplementary Fig. S5). The batch and cell type labels of the perturbed *CD14+* monocytes were highlighted to facilitate a detailed evaluation of scCobra and other methods in mitigating over-correction.

Supplementary Fig. S5: UMAP visualization of *CD14+* Monocytes and Perturbed *CD14+* Monocytes from the simulated immune dataset. a. Cells are colored by batch, demonstrating that perturbed *CD14+* Monocytes originate exclusively from the 10X batch. **b.** Cells are colored by cell type. This figure shows that perturbed cells are specific to the 10X batch, highlighting their batch-dependent origin.

After performing batch correction, we compared scCobra with Seurat, Harmony, Scanorama, scVI, scDML, and scDREAMER on the perturbed *CD14+* monocyte dataset. The corrected data were then visualized using UMAP, as shown in Supplementary Fig. S6b. The results showed that Seurat, Harmony, scVI, scDML, and scDREAMER failed to adequately separate perturbed *CD14+* monocytes from their normal counterparts, resulting in significant aggregation. In contrast, both Scanorama and scCobra successfully distinguished the two populations. However, Scanorama exhibited a tendency toward under-correction, as it failed to properly integrate other cell types, such as *CD4+* T cells, *CD8+* T cells, and *CD14+* monocytes, compromising overall data coherence. The UMAP visualization of scCobra-corrected *CD14+* monocytes and their perturbed counterparts revealed a clear separation between the two cellular states (Supplementary Fig. S6c), also highlighting scCobra's ability to minimize over-correction. Marker genes such as *NUP35*, *HSP90AA1*, and *NUP214*, associated with the "Reactome influenza life cycle" pathway, were identified as robust differentiators between these states, further validating scCobra's capacity to preserve biologically meaningful variations.

Supplementary Fig. S6 scCobra is capable of distinguishing perturbed *CD14+* Monocytes from their unperturbed counterparts. **b**, Batch correction methods (Seurat, Harmony, Scanorama, scVI, scDML, scDREAMER, and scCobra) were assessed on a simulated immune scRNA-seq dataset with batch effects and biological differences introduced by perturbing genes associated with the "Reactome influenza life cycle" Reactome pathway. The UMAP visualization is colored by cell type. The results demonstrate that scCobra effectively distinguishes perturbed *CD14+* Monocytes from their unperturbed counterparts during batch integration, with significantly lower levels of over-correction compared to other methods. **c**, Enlarged UMAP plots clearly show that scCobra effectively separates normal *CD14+* Monocytes from perturbed *CD14+* Monocytes. This distinction is highlighted by specific markers such as *NUP35*, *HSP90AA1*, *NUP214* demonstrating that differential markers can reliably distinguish perturbed cells from normal cells. This figure underscores scCobra's ability to maintain critical biological variation after batch correction.

To complement these qualitative findings, we quantified the extent of over-correction for all methods using the normalized over-correction score (Supplementary Fig. S6e). Across all cell types, scDML exhibited the highest degree of over-correction (0.251), followed by Seurat, Harmony, scVI, and scDREAMER, which displayed varying levels of over-correction. In contrast, both Scanorama (0.06) and scCobra (0.073) achieved lower over-correction scores, indicating their relative effectiveness in mitigating over-correction risk. Focusing specifically on *CD14+* monocytes and their perturbed counterparts, Seurat, Harmony, scVI, scDML, and scDREAMER all had over-correction scores exceeding 0.4. By comparison, Scanorama (0.26) and scCobra (0.31) exhibited significantly lower scores. However, the UMAP visualizations (Supplementary Fig. S6b) revealed that Scanorama's tendency toward under-correction compromised its ability to adequately separate cellular populations. scCobra, on the other hand, achieved a balance between minimizing over-correction and maintaining robust integration, preserving biologically meaningful distinctions between perturbed and normal cells.

Supplementary Fig. S6 scCobra is capable of distinguishing perturbed *CD14+* Monocytes from their unperturbed counterparts. e, Quantitative over-correction analysis on all cell types and specifically for *CD14+* Monocytes and perturbed *CD14+* Monocytes. Higher values indicate a greater degree of over-correction. The results demonstrate that scCobra significantly reduces the risk of over-correction compared to existing methods, as indicated by its lower over-correction scores.

Overall, our results on *CD4+* T cells and *CD14+* monocytes collectively validate scCobra's superior ability to minimize over-correction risk while ensuring accurate and biologically coherent integration.

We have updated Fig. 3 and added Supplementary Fig. S5, S6 accordingly. Additionally, we have made revisions to the "*scCobra minimizes over-correction risks in batch correction*" section of the manuscript.

C4-1: The GOBP analysis in the COVID-19 dataset requires further explanation: Please provide more details on how the GOBP analysis was performed on raw data.

Response: Thank you for your question. In the GOBP analysis of the COVID-19 dataset, we focused on identifying differences between healthy and disease samples. For the raw data, we first combined three healthy samples (AP1, AP2, and AP3) with three disease samples (BGCV01_CV0902, BGCV01_CV0904, and BGCV02_CV0902) and performed log normalization on the combined dataset, followed by the selection of highly variable genes.

We then extracted *CD4+* T cells from all samples and performed differential gene expression analysis using Omicverse⁷. Specifically, *CD4+* T cells from the healthy samples served as the control group, while those from the disease samples were treated as the treatment group. Significantly upregulated genes identified through this analysis were used for GOBP enrichment analysis to highlight the top 10 enriched pathways.

It is important to note that, since all batch correction methods had already preprocessed the original data, no additional preprocessing was applied to the corrected gene expression data before performing the differential analysis.

We have included detailed descriptions in the Methods section "*GOBP enrichment analysis*" explaining the procedures used to perform the GOBP enrichment analysis.

C4-2: Clarify why the "Defense response to Virus" pathway was spotted on here in COVID-19 data, when it was verified in the perturbed human immune dataset but not in COVID-19 data. Why the shared terms found by other methods (e.g., "Neutrophil activation involved in immune response" by scVI and Seurat, "cytokine-mediated signaling pathway" by Harmony) could not be used to explain the batch effect correction performance?

Response: Thank you for your thoughtful observation. The GOBP term "Defense Response to Virus" was selected as an illustrative example because it is captured in the raw data (without correction) and is strongly associated with COVID-19⁸. Ideally, if batch correction does not result in over-correction, this pathway should be retained after correction. However, this term was lost following correction with methods like Seurat, Harmony, Scanorama, and scVI, suggesting over-correction occurred with these approaches.

You are correct that other pathways, such as "Neutrophil activation involved in immune response"⁹ and "Cytokine-mediated signaling pathway"¹⁰ could also be associated with COVID-19. In fact, the most significant enriched pathways identified by scCobra, including "Cellular Response to Type I Interferon"¹¹ and "Viral Genome Replication"¹² are also relevant to COVID-19 pathology. The discussion of "Defense Response to Virus" was intended as a simple demonstration of the over-correction risk, not as comprehensive evidence.

We agree that while this example illustrates over-correction risk, it is not sufficient on its own to support the broader conclusions. To address this, we have now performed a more quantitative and systematic analysis of over-correction risk. For detailed results and discussions, please refer to our response to Reviewer #2 C3.

We have made revisions to the "*scCobra minimizes over-correction risks in batch correction*" section of the manuscript to clarify it more effectively.

C4-3: Explain why raw data is used as a ground truth, given potential batch effects in raw data. It's not convincing to use raw data as ground truth to validate scCobra's superiority in batch effect correction. Instead, employing a use case with known ground truth, such as simulated perturbed 10X data, would provide a more reliable benchmark.

Response: Thank you for your insightful comment. We would like to clarify that raw data was not used as ground truth to validate scCobra's performance. Instead, it served to demonstrate that certain biological signals, such as the "Defense Response to Virus" pathway, can be identified in the uncorrected data. However, many existing methods failed to preserve these signals post-correction, suggesting over-correction. In contrast, scCobra maintained these signals, highlighting its robustness in batch correction.

We agree with you that a simulated perturbation with known truth would be better to validate the batch correction performance of scCobra. In fact, we have already performed a simulation using *CD4+* T cells, where scCobra demonstrated its ability to accurately distinguish perturbed *CD4+* T cells from their unperturbed counterparts based on UMAP visualization. Additionally, the over-correction score confirmed that scCobra effectively reduces the risk of over-correction. To further enhance the results, we extended our simulations to another cell type, *CD14+* monocytes, with a different perturbation (For further details, please refer to the response to Reviewer #2 C3). This additional analysis further validated scCobra's superior performance in balancing batch effect removal while preserving meaningful biological signals, as compared to existing methods.

We appreciate your question here, as it prompted us to reflect more deeply on why, if raw data can identify reasonable biological signals, we cannot directly use raw gene expression for downstream analysis. Through two experiments, we demonstrated that raw gene expression contains significant batch noise, which can obscure or distort true biological differences. By applying batch correction, scCobra ensures that downstream analyses, such as differential gene expression, are based on more reliable and biologically coherent data. This further supports our argument that batch-corrected gene expression should be used for critical tasks to achieve accurate and meaningful results.

(1) Corrected gene expression can better reveal real biological differences

We analyzed the liver cancer dataset "GSE149614" focusing on cancerous tissue (HCC01T) and normal tissue samples (HCC03N). The UMAP visualization clearly showed a strong batch effect between these two sample types. We then applied batch correction in the original space using several methods, including Seurat, Harmony, Scanorama, scVI, and scCobra. As shown in the UMAP results (Fig. 3d), all methods successfully integrated normal and cancer samples.

Using the Omicverse toolkit⁷, we conducted differential gene expression analysis on both the raw and batch-corrected gene expression data (from scCobra and the benchmarking methods). The differentially expressed genes (DEGs) were then subjected to survival analysis following the Scissors⁶ pipeline. This method, utilizing bulk TCGA data, assesses the significance of DEGs in stratifying patients into high-risk and low-risk groups. The survival analysis results revealed that DEGs identified using scCobra and scVI were more effective in distinguishing healthy individuals from patients. This was evidenced by significantly lower survival-adjusted P of 4.1×10^{-21} and 1.1×10^{-19} , respectively (Fig. 3e). In contrast, the raw data resulted in the highest survival-adjusted P (1.0×10^{-8}), highlighting how batch effect signals can significantly compromise the reliability of differential analysis. Therefore, eliminating batch effects is essential for obtaining accurate and meaningful results. To demonstrate the robustness of our findings, we also performed differential gene screening under varying cutoff thresholds and conducted survival analyses based on the results (Supplementary Fig. S7). These analyses, consistent across different thresholds, further confirm the reliability and stability of our conclusions.

Fig. 3: scCobra Can Minimize Over-correction Risk. d, The UMAP visualizations of the liver cancer dataset, integrated using different methods and colored by cell type, demonstrate the batch correction capabilities of each approach. Compared to the raw data, Seurat, Harmony, Scanorama, scVI, and scCobra all effectively removed batch effects between healthy and diseased samples. **e,** Survival analysis results using differential gene sets obtained from corrected gene expression with scCobra and benchmarking methods. Blue represents high-risk groups, and orange represents low-risk groups, with the x-axis indicating survival days and the y-axis representing the survival proportion. The analysis demonstrates that scCobra-corrected gene expression eliminates batch noise and identifies meaningful differential genes that most effectively stratify patients into high-risk and low-risk groups, improving survival stratification compared to raw and corrected gene expressions from existing methods.

(2) Batch-corrected gene expression reduces batch noise in the raw data.

We also performed differential gene analysis between two healthy samples, HCC03N and HCC04N. Since both samples are from healthy individuals, we expect there should be no true differential genes between these two samples within same condition. However, using the raw gene expression data, we identified 470 differential genes, which is likely due to batch noise in the raw data, with a significant portion of the differential genes being driven by batch effects. After applying batch correction, the batch noise was reduced, and the number of differential genes between the two healthy samples decreased. Specifically, scCobra identified 30 differential genes, scVI identified 66 genes, while Harmony identified 40. The number of differential genes identified by Seurat and Scanorama also decrease (Supplementary Fig. S3). Our results further emphasize the importance of using corrected gene expression data to reduce batch noise, which is crucial for accurately detecting genuine biological signals.

Supplementary Fig. S3: scCobra effectively eliminates spurious differential signals caused by batch effects. The figure shows the differences in the number of differentially expressed genes (DEGs) between samples under the same condition, comparing raw gene expression with corrected gene expression using scCobra and other methods. The results demonstrate that scCobra significantly reduces the number of DEGs driven by batch effects, outperforming other methods in eliminating batch-induced noise.

We have replaced the enrichment results from the previous version with the survival analysis results in the main manuscript (Fig. 3) to provide a clearer and more intuitive demonstration of over-correction, thereby enhancing the evaluation of scCobra's performance compared to other methods. Additionally, we have included Supplementary Fig. S3 and Supplementary Fig. S7 to further support our conclusions. Revisions have also been made to the "*scCobra minimizes over-correction risks in batch correction*" section of the manuscript. Furthermore, the enrichment analysis is retained in Supplementary Fig. S8 for reference.

C5: The UMAP visualizations in Figure 4c compare the batch correction performance of various methods. Given that the simulated data is well-clustered by cell type, simulating single-cell data with continuous underlying structures might better demonstrate scCobra's strengths in batch effect correction.

Response: Thank you for your thoughtful suggestion. We agree that simulating single-cell data with continuous underlying structures can provide valuable insights into batch correction performance. However, scCobra generates new data by learning batch information from the original data. Since the original data we used was generated by Splatter¹³, which is relatively simple and lacks complex continuous structures, it is challenging to produce data with continuous underlying structures in this context. To address this, we extended our analysis to include more complex real-world datasets with continuous gene expression profiles, as they naturally exhibit such structures. These datasets provide a more biologically relevant benchmark for evaluating scCobra's generative capabilities and strengths in batch effect correction.

We have updated the manuscript to include these analyses, along with UMAP visualizations (Fig. 5) and quantitative results (see Supplementary Fig. S9). This inclusion highlights scCobra's ability to accurately correct batch effects while preserving the underlying biological structures inherent in continuous gene expression profiles. We believe this addresses your concern and further strengthens the evaluation of scCobra's performance.

Following the batch generation strategy, as illustrated in Fig. 5a, we first integrated data from three batches (indrop1, smartseq2, and celseq) and used the Domain-Specific Batch Normalization layer to map the embeddings of smartseq2 into the spaces of indrop1, smartseq2, and celseq, thereby simulating batch effects (Fig. 5b). We then applied scCobra and other benchmarking methods to integrate the generated batch-effect data (Fig. 5c). Although our generated dataset contains batch effects, its source remains consistent, and the cell labels are unaffected by the addition of batch information. This consistency makes it highly effective for evaluating batch correction methods.

Fig. 5 scCobra's Efficacy in Simulating Single-Cell Data with Batch Effects. **a**, Illustration of scCobra's capacity to generate single-cell RNA-seq data simulations that include batch effects, showcasing the adaptability and application of the scCobra model. **b**, Through UMAP visualization, this section demonstrates the effectiveness of scCobra in removing batch effects from data, as well as showcasing real single-cell data re-generated with batch labels, highlighting its practical utility in real-world scenarios. **c**, Features UMAP visualizations of simulated data after batch correction, the first row shows the integration of batches, while the second row demonstrates the clustering of cells of the same type, with these panels designed to illustrate the efficiency of various methods in correcting batch effects within datasets simulated by scCobra. However, this simulation acts as a benchmarking to evaluate the performance of other batch correction tools such as Seurat, Harmony, Scanorama, scVI, scDML and scDREAMER, as shown in panel c.

In summary, scCobra can generate data with batch effect by learning from the batch information in existing data. This approach guarantees uniform cell type annotations across the simulated batches, effectively removing potential annotation inconsistencies. It allows for an unbiased evaluation of batch correction techniques.

We have updated Fig. 5 and revised the section "*scCobra enables simulation of single-cell data with batch effects*" in our manuscript. Additionally, we have moved the results using the simulation dataset to Supplementary Fig. S10 for reference.

C6: In the online label transfer section, if the model is jointly trained on both reference and query datasets (as stated by “We performed joint training of the raw data from the pancreas reference dataset and query dataset, facilitating the transfer of labels from the pancreas reference dataset to the query datasets.”), the metrics may reflect training accuracy rather than generalization ability. To better assess generalization, it would be helpful to train the model only on reference data and test annotation accuracy on the query datasets.

Response: Thank you for pointing out this important clarification. We apologize for any confusion caused by the original description. In our label transfer workflow, the reference and query datasets are independent, ensuring that the evaluation metrics reflect generalization rather than training accuracy.

Specifically, the implementation involves two stages: first, integrating a multi-batch dataset to construct a reference atlas and training a KNN classifier based on the true cell labels in the reference dataset. Second, the query dataset is processed using the pre-trained encoder to remove batch effects in the latent space, mapping the query dataset to the constructed reference atlas. The pre-trained KNN classifier is then applied to annotate the query dataset, ensuring that the model generalizes effectively to unseen data.

For example, in our pancreas dataset, which consisted of 16,400 cells from eight batches, scCobra was used to integrate the data and construct a reference atlas. The pre-trained encoder was saved, and a KNN classifier was trained using the true cell labels from the reference dataset. The query dataset (comprising three batches) was then encoded and mapped to the reference atlas, where the KNN classifier successfully predicted cell labels.

This independent handling of reference and query datasets ensures that the evaluation focuses on generalization rather than training accuracy. Additionally, scCobra demonstrated superior annotation accuracy compared to other benchmark methods, further highlighting its effectiveness in label transfer tasks.

To provide greater clarity on our training strategy, we have updated the description of the implementation of label transfer in the Methods section “*Online label transfer*” of the manuscript.

MC1: As scCobra is a deep learning-based model, a more detailed description of its architecture is necessary. Please clarify the number of hidden layers in the encoder, decoder, and discriminators, as well as the number of neurons in each layer, the dimension of latent factors, the temperature

coefficient in contrastive learning loss, and the trade-off parameters lambda used in the loss functions.

Response Thank you for your insightful comments and suggestions. To address the comment regarding the detailed architecture of scCobra, we have provided a comprehensive description of the model's components and parameters. The details are summarized in the following tables:

Supplementary Table 2: Model architecture parameters

Module	Dimension
Encoder (E)	2000, 1024, 10
Decoder (D)	10, 2000
Fixed Encoder (E_f)	2000, 1024, 10
Domain Discriminator (Dis^d)	10, batch number
Cell level Contrastive head (Dis^c)	10, 10
Cluster level Contrastive head (Dis^l)	10, cluster number
Discriminator head (Dis^g)	10, 1

Supplementary Table 3: Loss function parameters and trade-off parameters

Parameter	Value
λ	1
λ_{KL}	1
λ_{vae}	1
Temperature of Cell level Contrastive loss	0.1
Temperature of Cluster level Contrastive loss	0.5

These tables comprehensively outline the number of hidden layers in the encoder, decoder, and discriminators, the number of neurons in each layer, the latent factor dimensions, the temperature coefficients for contrastive learning, and the trade-off parameters (λ) used in the loss functions.

To enhance the accessibility of this information, we have added these tables to Supplementary Table 5 and Supplementary Table 6. We hope this addresses your concern, and we are happy to provide further clarifications if needed.

We have included detailed descriptions of scCobra's model architecture parameters, loss function parameters, and trade-off parameters in the supplementary material and referenced them in the Method section "*The scCobra pipeline*".

MC2: The datasets used in the results should be clarified clearly. If they're scRNA-seq data, please use "scRNA-seq data" instead of "single-cell datasets".

Response: Thank you for your insightful suggestion. In line with your recommendation, we have thoroughly revised the manuscript to replace all instances of "single-cell datasets" with "scRNA-

seq data" where applicable. This modification enhances the precision and clarity of our language, ensuring that the terminology accurately reflects the specific type of data analyzed in our study. Additionally, this adjustment aligns with standard practices and terminology commonly used in the field, thereby improving the overall readability and consistency of the manuscript.

MC3: Explain how the over-correction score is computed and normalized. What do negative values in the results signify?

Response: We sincerely thank you for your thoughtful questions. We used SCALEX⁵ defined method to evaluate the over-correction, referred to as the over-correction score. This score quantifies the severity of over-correction by calculating the percentage of inconsistencies in cell types within the neighborhood of each cell. Specifically, the score is computed across all cells by averaging the frequency of k -nearest neighboring cells with distinct cell types for each cell i . The formula is as follows:

$$\text{over_correction_score} = 1 - \frac{1}{n \cdot k} \sum_{i=1}^n \sum_{j=1}^k I(\text{cell_type}_i, \text{cell_type}_j)$$

Where:

- n : Total number of cells.
- k : Number of k -nearest neighbors for each cell
- cell_type_i : cell type of cell i
- cell_type_j : cell type of neighboring cell j
- $I(\text{cell_type}_i, \text{cell_type}_j)$: Indicator function defined as:

$$I(\text{cell_type}_i, \text{cell_type}_j) = \begin{cases} 1, & \text{if } \text{cell_type}_i = \text{cell_type}_j \\ 0, & \text{if } \text{cell_type}_i \neq \text{cell_type}_j \end{cases}$$

The over-correction score is a negative metric, meaning that higher scores indicate more severe mixing of cell types and, consequently, a greater extent of over-correction.

Normalization and Interpretation

To normalize the results, we subtracted the over-correction score computed on the raw data from the over-correction scores of all methods. This approach ensures that the metric evaluates over-correction relative to the baseline (raw data).

On Negative Values

In the original manuscript, we anticipated that negative values in the normalized over-correction score would indicate under-correction. However, further analysis revealed that these negative values might result from randomness introduced during the calculation process. Specifically, in the Python implementation, random sampling is performed, which inherently introduces a degree of stochasticity.

Since the raw data undergoes no batch correction, its over-correction score is theoretically the lowest. However, minor fluctuations caused by random sampling could lead to scenarios where the over-correction score of certain methods (e.g., Scanorama) becomes slightly lower than that of the raw data. When normalizing by subtracting the raw data's over-correction score, this may result in negative values. For example, the reported value of -0.02 in the original manuscript is very close to 0 and is most likely caused by random fluctuations.

To ensure the accuracy and robustness of the results, we compute the over-correction score using five different random seeds and average the results. This approach effectively reduces the impact of random fluctuations and provides a more reliable estimate of the over-correction score.

Additionally, we have updated both the figures and the textual descriptions in the manuscript to accurately reflect this finding, including the over-correction benchmarking in the '*scCobra minimizes over-correction risks in batch correction*' section, and have provided a more detailed description of this part in the Methods section "*Over-correction evaluation*".

MC4: In Fig. 2b, the expression values of *IR27* and *TLR9* seem to only contain two values, which is uncommon for scRNA-seq data. Please verify the expression data for these genes and explain why they were chosen for demonstration. Are these genes perturbed in the data, or are they differentially expressed across *CD4+* T cell states?

Response: Thank you for your question. The observed two-value pattern for genes *IR27* and *TLR9* arises from a combination of their expression characteristics in the raw data and the visualization parameters used. These genes have zero expression in the majority of cells, which, when perturbed, results in only two distinct visualized values—zero and the perturbed expression level. This behavior is typical for genes with sparse expression in scRNA-seq data.

We selected *IR27*, *TLR9*, and *HSP90AA1* for demonstration because they are differentially expressed across normal and perturbed *CD4+* T cells in the disturbance experiments. While *IR27* and *TLR9* exhibit binary-like expression patterns due to sparsity, *HSP90AA1* shows multiple distinct expression levels, providing a broader dynamic range for visualization (see Fig. 3b and Supplementary Fig. S4d). These genes were chosen as they highlight the capability of scCobra to separate perturbed and normal cell states, even for sparsely expressed genes.

Fig. 3 scCobra Can Minimize Over-correction Risk. b. Enlarged UMAP plots clearly show that scCobra effectively separates normal *CD4+* T cells from perturbed *CD4+* T cells. This distinction is highlighted by specific markers such as *TLR9*, *HSP90AA1*, and *IL27*, demonstrating that differential markers can reliably distinguish perturbed cells from normal cells. This figure underscores scCobra's ability to maintain critical biological variation during batch correction.

Supplementary Fig. S4 scCobra is capable of distinguishing perturbed *CD4+* T cells from their unperturbed counterparts. **d**, The x-axis represents gene expression levels, while the y-axis indicates the frequency of cells expressing the corresponding levels. This figure demonstrates that not all genes exhibit binary expression patterns, with some showing a range of expression levels across cells.

Additionally, in experiments with *CD14+* Monocytes, we observed similar expression patterns for perturbed genes, as shown in Supplementary Fig. S6c and Supplementary Fig. S6d. These examples further illustrate scCobra's ability to preserve biological differences while minimizing over-correction.

Supplementary Fig. S6 scCobra is capable of distinguishing perturbed *CD14+* Monocytes from their unperturbed counterparts. **c**, Enlarged UMAP plots clearly show that scCobra effectively separates normal *CD14+* Monocytes from perturbed *CD14+* Monocytes. This distinction is highlighted by specific markers such as *NUP35*, *HSP90AA1*, *NUP214* demonstrating that differential markers can reliably distinguish perturbed cells from normal cells. This figure underscores scCobra's ability to maintain critical biological variation during batch correction. **d**, The x-axis represents gene expression levels, while the y-axis indicates the frequency of cells expressing the corresponding levels. This figure demonstrates that not all genes exhibit binary expression patterns, with some showing a range of expression levels across cells.

We have added Supplementary Fig. S4d and Supplementary Fig. S6d to address your comment. Additionally, we have made revisions to the "*scCobra* minimizes over-correction risks in batch correction" section of the manuscript.

MC5: The positioning of OD immature and OD mature cells in Fig. 3d does not appear as close as suggested by the manuscript. Please review the figure and corresponding statements for consistency.

Response: Thank you for pointing out the discrepancy. Upon review, we realized that our description in the manuscript was inaccurate. The correct observation is that "Newly formed ODs" and "Immature ODs" are positioned closer together, whereas "Mature ODs" are relatively farther apart. We have revised the corresponding text in the manuscript to accurately reflect the spatial relationships depicted in Fig. 3d. Additionally, we ensured consistency between the figure and its description to avoid any potential confusion. We appreciate your attention to this detail.

Fig. 4 Multimodal batch correction. **d**, the UMAP visualization results of the single-cell spatial dataset before and after batch correction. In the first row, the colors represent the MERFISH data and RNA-seq data respectively, while in the second row, the colors represent the cell types.

We have revised the textual descriptions in the section "*scCobra enables multi-omic batch correction.*"

MC6: Provide more details on the simulated datasets generated by Splatter. In Figure 4b, although six batches are simulated, only two batches are used in Figure 3c. Consider omitting unnecessary batches from the figures. Additionally, clarify the goal of batches 3-6 as they appear to be clustered by cell type without batch effects.

Response: Thank you for highlighting these issues. Splatter¹³ simulates data by generating distinct clusters based on a negative binomial distribution. By varying random seeds and parameters, it produces datasets with batch effects, making it a versatile tool for evaluating batch correction methods. Regarding the comment: "Consider omitting unnecessary batches from the figures," we apologize for the misunderstanding. We have already removed redundant batches and retained only the most relevant ones for UMAP visualization. You also mentioned: "Additionally, clarify the goal of batches 3-6 as they appear to be clustered by cell type without batch effects" scCobra

generates new data with batch effects by learning batch information from the original data. To achieve this, it first integrates the original data. In Fig. 5b, we first present the results of scCobra's integration of the original data, followed by the generation of new data containing batch effects.

Beyond this, we acknowledge that the simulated dataset in the original manuscript was overly simplistic and insufficient to represent the challenges of real-world batch correction, so we replaced the simulated dataset with a real-world pancreas dataset and further generated a more complex dataset with enhanced batch effects (see response to Reviewer #2 C5).

We have updated Fig. 5 and revised the section "*scCobra enables simulation of single-cell data with batch effects*" in our manuscript. Additionally, we have moved the results using the simulation dataset to Supplementary Fig. S10 for reference.

MC7: The figure descriptions should focus on describing the contents rather than interpreting the results. For example, in Figure 2b, explain what "normalized over-correction score" means. Is a higher score better? in Figure 4b Why are batches 1 and 2 separated from batches 3-6 in the plots? What's the metric used in figure 5b-e to measure accuracy?

Response: We appreciate your feedback and have ensured that the figure descriptions focus on describing the content rather than interpreting the results.

The normalized over-correction score is a normalized metric used to evaluate the extent of over-correction (refer to Reviewer #2 MC3), higher scores indicate a greater degree of over-correction. For details, please refer to the updated caption of Fig. 3c.

The metric used in Fig. 6b-e to measure accuracy is the accuracy (ACC) calculated for each cell type. Specifically, we evaluate whether the predicted cell labels match the true labels for each individual cell. The accuracy for a given cell type is then determined by the proportion of correctly predicted labels relative to the total number of cells in that type. You can refer to Fig. 6b-e to check our revised figure caption.

For the question "*In Fig. 4b, why are batches 1 and 2 separated from batches 3-6 in the plots?*", that because we just use two batches' data to generate new data. However, the simulated dataset was overly simplistic and insufficient to generate realistic and complex batch effects. Therefore, we have replaced the simulated dataset with a real dataset to better evaluate the method under more authentic conditions. We have also replaced the simulated dataset with a real dataset, and both the figure and its caption have been updated according to your suggestions (refer to Reviewer #2 MC6, Reviewer #2 C5).

We have made revisions to the figure captions you mentioned, focusing on describing the content rather than interpreting the results.

MC8: Annotate data source in figure 3a and figure 3d for better understanding.

Response: Thank you for your valuable suggestion. We have meticulously clarified the data sources in the manuscript. Specifically, Fig. 4a utilizes multi-omics datasets that combine single-cell RNA sequencing (scRNA-seq) and single-cell ATAC sequencing (scATAC-seq). These datasets are the Cell ranger example data provided by 10X and are sourced from human peripheral

blood mononuclear cells (PBMC) (<https://www.10xgenomics.com/datasets/10-k-pbm-cs-from-a-healthy-donor-v-3-chemistry-3-standard-3-0-0>, <https://www.10xgenomics.com/datasets/10-k-peripheral-blood-mononuclear-cells-pbm-cs-from-a-healthy-donor-1-standard-1-2-0>). Meanwhile, Fig. 4d is based on spatial transcriptomics data, which pertains to the spatial transcriptome data of the mouse hypothalamus region¹⁴.

This detailed information has been comprehensively included in the "Data Availability" section to ensure maximum clarity and offer a more in-depth context for enhanced understanding. We sincerely appreciate your feedback and your meticulous attention to detail, as it has significantly contributed to the refinement and comprehensibility of our research.

We have added the data annotation in the section "*scCobra enables multi-omic batch correction*".

MC9: In the loss function of domain discriminator and GAN, should x_i and b_j use the same index i ?

Response: Thank you for your insightful comment. You are right that in the loss function for the domain discriminator and GAN, the indices should align to maintain consistency between cells and their corresponding batches. Specifically, single cell expression (x_i) should be paired with its batch label (b_i) to accurately reflect this relationship. We have corrected the notation in the manuscript to resolve this issue and ensure clarity in the mathematical representation. Thank you for catching this detail, which has helped us refine the accuracy of our explanation.

Accordingly, we have revised the formulas and descriptions related to batch labels in the original manuscript.

For domain discriminator (Dis^d), Its optimization goal is defined as:

$$L_d = \min_{\omega} \sum_{i=1}^N CE \left(Dis_{\omega}^d (E(x_i)), b_i \right)$$

Where E represents the encoder, Dis^d represents the batch label discriminator, x_i represents a single cell, b_i represents the batch label, CE represents the cross-entropy loss, N is the number of cells in the single-cell dataset.

After optimizing the domain discriminator, we can train the generator to maximize the domain discriminator's loss, effectively removing batch-specific information from the latent space. The optimization objective is defined as follow:

$$L_G = \max_{\rho} \sum_{i=1}^N CE \left(Dis^d \left(E_{\rho}(x_i) \right), b_i \right)$$

Where E represents the encoder, D represents the decoder, Dis^d represents the batch label discriminator, x_i represents a single cell, b_i represents the batch label, CE represents the cross-entropy loss, N is the number of cells in the single-cell dataset

We have revised the Methods section "*The scCobra pipeline*" of the manuscript to update the formula related to the batch label.

MC10: What's the difference between contrastive learning at the single-cell level and the cluster level? Both l_{cell} and $l_{cluster}$ use x_i and x_i' at single-cell level.

Response: Thank you for your insightful question. The key difference between contrastive learning at the single-cell level and the cluster level lies in their granularity and optimization focus, as clearly illustrated in the diagram (Fig. 1). Contrastive clustering¹⁵, which inspired our approach, effectively combines instance-level and cluster-level contrastive learning to generate meaningful image representations while simultaneously enabling clustering. Building on this concept, we integrated both levels of contrastive learning into the single-cell VAE-GAN framework, resulting in the development of scCobra. Below, we provide a clearer explanation of the differences between these two types of contrastive learning and how they are utilized in scCobra.

Cell-level contrastive learning (l_{cell}):

At the single-cell level, the focus is on aligning the representations of the same cell under different views (e.g., input and reconstruction). In the diagram, these are represented as positive pairs (connected by "pull" arrows), while all other cells in the minibatch are treated as negative samples (pushed away, as shown by "push" arrows). The goal is to ensure that the latent representations of the same cell are close, while separating them from representations of other cells.

Cluster-level contrastive learning ($l_{cluster}$):

At the cluster level, the focus shifts to aligning representations of clusters. The cluster-level contrastive head aggregates the outputs of cells to form cluster representations. Positive pairs are representations of the same cluster, while negative pairs are representations from different clusters, as depicted in the diagram. The "pull" and "push" arrows indicate the optimization objective of pulling representations within the same cluster closer and pushing representations from different clusters apart.

And for your second question why both l_{cell} and $l_{cluster}$ use x_i and x_i' at single-cell level,

For cluster-level contrastive learning, the cluster-level contrastive head includes a SoftMax classifier that outputs the probabilities of each cell x_i and x_i' belonging to M clusters. This results in a $2K \times M$ matrix, where K is the number of cells in the minibatch, and $2K$ accounts for both the original input and the reconstructed output.

To facilitate cluster-level contrastive learning, we first transpose this $2K \times M$ matrix to obtain a $M \times 2N$ matrix. From this, we extract two separate $M \times K$ matrices:

1. One represents the cluster-level embeddings derived from the original inputs.
2. The other represents the cluster-level embeddings derived from the reconstructed outputs.

Using these two $M \times K$ matrices, we can perform contrastive learning at the cluster level. Positive pairs are formed between corresponding clusters (e.g., Cluster j from the original input and Cluster j from the reconstructed output), while clusters from different groups are treated as negatives. This

approach enables us to align cluster representations effectively, ensuring consistent structure across the latent space.

In summary, cell-level contrastive learning emphasizes fine-grained alignment of single-cell representations, while cluster-level contrastive learning promotes global structure by distinguishing between clusters. Together, these two levels of contrastive learning ensure that scCobra captures both detailed single-cell characteristics and meaningful cluster-level relationships.

To better clarify how we perform cell-level and cluster-level contrastive learning, we have added a dedicated section “*Cell-level contrastive learning and cluster-level contrastive learning*” in the Methods part of the manuscript to explain these two aspects of the scCobra methodology in detail.

MC11: Figure 7a, it seems like the cells are colored by cell type not batch. If so please correct titles.

Response: Thank you for pointing this out. You are correct that the cells in Fig. 7a were mistakenly colored by cell type instead of batch. We apologize for the oversight and have corrected the figure accordingly. The titles have also been updated to accurately reflect the content of the figure. We appreciate your careful review and attention to detail, which helped us improve the clarity and accuracy of our manuscript.

Fig. 7 Ablation results for all components. a, UMAP visualization results of model correction pancreas scRNA-seq dataset excluding contrastive learning, excluding GAN, excluding AdaBN and complete scCobra, colored by cell type.

We have revised the Fig. 7a’s annotation.

MC12: Correct typos in manuscript, e.g., “minimizes” rather “mininizes”, “modality” rather “modal”.

Response: Thank you for pointing this out. We have carefully reviewed the entire manuscript and corrected all typographical errors, including replacing “mininizes” with “minimizes” and “modal” with “modality.” Additionally, we conducted a thorough check for other potential spelling and grammatical errors throughout the text to ensure clarity and accuracy. We appreciate your attention to detail, which helped us improve the quality of the manuscript.

Reviewer #3

General Comments: In the field of single cell data analysis, many methods have been developed to correct batch effects. However, these methods are limited by the prior assumptions of gene distribution and have the problem of over-correction. This study proposes a new method for batch correction, based on the neural network architecture of VAE-GAN and the strategy of contrastive learning. This method can not only solve the problem of batch over-correction, but also realize multimodal batch correction, simulating the batch effects, and label transfer.

Overall, this is an interesting study and this new approach could be a useful tool and a great complementary to the field. The work is technically strong and the manuscript is well-written and presented. I have a few questions to be addressed that are listed below.

Response: Thank you very much for your kind and positive comments. We greatly appreciate your recognition of the originality and significance of scCobra, as well as your acknowledgment of its contributions to batch correction and multimodal integration. Your feedback is highly motivating, and we are glad that you find our work both technically robust and well-presented. We have carefully considered your suggestions and made revisions to address the points you raised. We hope the updated manuscript meets your expectations and contributes significantly to the field of single-cell data analysis.

C1: Can the authors generate simulated datasets with more scenarios to further evaluate the method? For example, a dataset contains at least two batches with imbalanced cell subpopulation compositions (i.e., one cell group is specific to one of the batches), and a dataset has high batch complexity with nested sub-batches (i.e., one batch may contain three sub-batches).

Response: Thank you for your insightful suggestions. Upon reflection, we acknowledge that the simulated dataset in the original manuscript was overly simplistic and insufficient to represent the challenges of real-world batch correction. While this allows for basic simulations, the generated data lacks complexity and does not capture challenging scenarios such as imbalanced cell subpopulation compositions or high batch complexity with nested sub-batches. This limitation makes it difficult to comprehensively evaluate the batch generation capabilities of scCobra.

To address this issue, we replaced the simulated dataset with a real-world pancreas dataset and further generated a more complex dataset with enhanced batch effects. This dataset includes three batches (indrop1, smartseq2, and celseq) with pronounced batch effects, effectively showcasing the capability of scCobra in generating data with complex batch effects (see Fig. 5b).

Fig. 5 scCobra's Efficacy in Simulating scRNA-seq Data with Batch Effects. **b**, UMAP visualization demonstrates the effectiveness of scCobra in removing batch effects and highlights its practical utility in real-world scRNA-seq data by re-generating the data with batch labels.

The revised analysis focuses on these three batches, ensuring alignment with the core goals of the study (See Fig. 5c).

Fig. 5 scCobra's Efficacy in Simulating scRNA-seq Data with Batch Effects. **c**, Features UMAP visualizations of simulated data after batch correction, the first row shows the integration of batches, while the second row demonstrates the clustering of cells of the same type, with these panels designed to illustrate the efficiency of various methods in correcting batch effects within datasets simulated by scCobra. This simulation acts as a benchmarking to evaluate the performance of other batch correction tools such as Seurat, Harmony, Scanorama, scVI, scDML and scDREAMER, as shown in panel c.

By incorporating real-world data with clear batch effects, we addressed the limitations of the simulated datasets, making our results more robust and relevant. Thank you for your valuable feedback, which greatly improved this aspect of our manuscript.

We have updated Fig. 5 and revised the section "*scCobra enables simulation of single-cell data with batch effects*" in our manuscript. Additionally, we have moved the results using the simulation dataset to Supplementary Fig. S10 for reference.

C2: In the model architecture diagram, there are two encoders. The manuscript mentions that the parameters of encoder E_f are fixed and consistent with the parameters of encoder E . In the optimization process mentioned in the manuscript, the first step is to optimize the two discriminators, and the updated parameters are the parameters of the discriminator. The second step is to optimize the generator, at which time the parameters of the encoder and decoder are updated. I am wondering whether the parameters of encoder E and E_f are shared here? In addition, the contrast loss, reconstruction loss and KL loss are introduced in the third step. The gradient return of these losses will also affect the parameters of the encoder and decoder. Are the second and third step carried out successively? In general, VAE uses KL loss and reconstruction loss to return gradients when optimizing and updating parameters. Why does the manuscript optimize the second and third steps separately? Finally, is the reconstruction loss the MSE? It will be better if the authors can write the mathematical formula in more detail in the method.

Response: We sincerely thank you for the thoughtful and detailed comments.

First, regarding the relationship between E and E_f , their parameters are indeed shared. However, E_f does not receive gradients directly. Instead, at the beginning of each epoch, E_f updates its parameters by copying those of E , ensuring consistency between the two.

Second, you are correct that both the second and third steps are aimed at optimizing the encoder and decoder parameters. However, we would like to emphasize that multi-phase training is highly beneficial for facilitating the convergence of neural networks and it is a common approach to direct gradients more effectively to specific layers of the network¹⁶. So, in the second phase, we primarily optimize the generator (encoder and decoder) using the adversarial loss. In the third phase, we focus on optimizing the contrastive loss, KL divergence, and reconstruction loss. Finally, we perform an all-together training step to integrate all components and ensure model convergence.

Lastly, our reconstruction loss is implemented as a binary cross-entropy (BCE) loss. This is because, during preprocessing, all input values were normalized to the range of 0 to 1, making BCE loss a more suitable choice^{5,17}. Correspondingly, the reconstruction loss is:

$$L_{reconstruction} = \sum_{i=1}^N [-x_i \cdot \log(x'_i) - (1 - x_i) \cdot \log(1 - x'_i)]$$

This term is the Binary Cross-Entropy (BCE) loss, used to measure the difference between the original input x_i and the reconstructed output x'_i , N is the number of cells in the single-cell dataset. It reflects the quality of the model's reconstruction.

We have added descriptions in the Methods section “The scCobra pipeline” of the manuscript, addressing whether parameters are shared and the use of BCE as the reconstruction loss. Furthermore, we have updated the relevant formula in the main text to clarify the implementation of BCE and included additional references and explanations to support the rationale behind our training approach.

C3: In contrast loss, h_i in $[I]_{\text{cluster}}$ should be the representation of a cluster. How can we get the representation of the entire cluster from the representation of a cell in a cluster?

Response: Thank you for your question, it seems our explanation may have caused some misunderstanding. Here, h_i and h_i' represent the embeddings of the i -th cell's original input x_i and reconstructed output x_i' , obtained through E_f . These embeddings are not the representations of the i -th cluster, but rather correspond to individual cells. Next, I will explain how we derive the representation for an entire cluster.

For cluster-level contrastive learning, the cluster-level contrastive head includes a SoftMax classifier that outputs the probabilities of each cell x_i (h_i) and x_i' (h_i') belonging to M clusters. This results in a $2K \times M$ matrix, where K is the number of cells in the minibatch, and $2K$ accounts for both the original input and the reconstructed output.

To facilitate cluster-level contrastive learning, we first transpose this $2K \times M$ matrix to obtain a $M \times 2K$ matrix. From this, we extract two separate $M \times K$ matrices (**representing the embeddings for each cluster, as you asked**),

1. One represents the cluster-level embeddings derived from the original inputs.
2. The other represents the cluster-level embeddings derived from the reconstructed outputs.

Using these two $M \times K$ matrices, we can perform contrastive learning at the cluster level. Positive pairs are formed between corresponding clusters (e.g., Cluster j from the original input and Cluster j from the reconstructed output), while clusters from different groups are treated as negatives. This approach enables us to align cluster representations effectively, ensuring consistent structure across the latent space.

To better clarify how we perform cell-level and cluster-level contrastive learning, we have added a dedicated section “*Cell-level contrastive learning and cluster-level contrastive learning*” in the Methods part of the manuscript to explain these two aspects of the scCobra methodology in detail.

C4: Can you give a more specific formula interpretation for the BN block designed for the model? For example, how is the mean and variance of each batch determined? Is it to record the mean and variance of the original data or the mean and variance of the reconstructed data?

Response: Thank you for your question, we are happy to provide more details.

To address your first question:

For first question, how is the mean and variance of each batch determined?

The mean (μ_B) and variance (σ_B^2) of each batch are calculated directly from the original data (x_i) within the batch. For a batch with B cells, the mean μ_B and variance σ_B^2 are calculated as follows:

$$\mu_B = \frac{1}{B} \sum_{i=1}^B x_i, \quad \sigma_B^2 = \frac{1}{B} \sum_{i=1}^B (x_i - \mu_B)^2$$

Here, x_i refers to the original data for each cell in the batch.

For second question, is it to record the mean and variance of the original data or the reconstructed data?

Yes, the DSBN¹⁸ is designed to restore the batch information of the original data. Therefore, it records the mean and variance of the original data, not the reconstructed data. In scCobra, the output of the decoder is batch-invariant. To recover the true batch information, the DSBN utilizes gradients from the loss computed between the reconstructed data and the original data.

To provide better clarification of the BN block used, we have added detailed descriptions in the Methods section titled “*The scCobra pipeline*” of the manuscript.

MC1: The work only considers RNA and ATAC in multimodal batch integration. Is it applicable to other modalities such as RNA and Protein?

Response: Thank you for your question! In theory, scCobra can be extended to integrate other modalities, such as RNA and protein. However, scCobra relies on a shared encoder, which requires the selection of shared features. Common single-cell protein data, such as CITE-seq, typically contain only a few hundred features, making it challenging to select shared features and optimize the model effectively. Therefore, while it is theoretically feasible, practical applications would require addressing these limitations to achieve optimal performance.

We have added a description of this limitation in the *Discussion* section.

MC2: For the label transfer task, the authors mention that there is no need to retrain on new data, only pre-training on a reference data. Is there any requirement for the quality of the reference data, such as data volume, labels, etc.?

Response: We sincerely thank you for this thoughtful comment. Indeed, there are certain requirements for the reference data in the label transfer task. These requirements are as follows:

1. Tissue or organ consistency:

Label transfer requires that the query data comes from the same tissue type or organ as the reference atlas, ensuring that their expression patterns are closely aligned. Notably, cell annotations are not required for the query data.

2. Diverse batch inclusion:

Constructing the reference atlas requires multiple batches of data. scCobra uses a shared encoder across batches to create a unified reference atlas. Incorporating diverse batch types, such as data from 10X and Smart-seq platforms, is crucial to enhance the model's generalization capability. However, the total data volume is less critical. For example, in our integrated pancreas dataset, the reference atlas consisted of 16,400 cells from eight batches, while the query dataset contained approximately 24,000 cells from three batches. Despite this difference, scCobra successfully transferred cell labels to the query dataset without retraining.

We have included a brief description of the requirements for label transfer in the *Discussion* section.

MC3: Some text on the figures are too small, particularly the figure legends.

Response: Thank you for bringing this to our attention. We have addressed this issue by increasing the font size of the text and figure legends in all relevant figures. This adjustment ensures that the text is legible, and the figures are more accessible to readers.

References

1. Luecken, M.D. et al. Benchmarking atlas-level data integration in single-cell genomics. *Nature Methods* **19**, 41-50 (2022).
2. Yu, X., Xu, X., Zhang, J. & Li, X. Batch alignment of single-cell transcriptomics data using deep metric learning. *Nature communications* **14**, 960 (2023).
3. Shree, A., Pavan, M.K. & Zafar, H. scDREAMER for atlas-level integration of single-cell datasets using deep generative model paired with adversarial classifier. *Nature Communications* **14**, 7781 (2023).
4. Lu, Y. et al. A single-cell atlas of the multicellular ecosystem of primary and metastatic hepatocellular carcinoma. *Nature communications* **13**, 4594 (2022).
5. Xiong, L. et al. Online single-cell data integration through projecting heterogeneous datasets into a common cell-embedding space. *Nature Communications* **13**, 6118 (2022).
6. Sun, D. et al. Identifying phenotype-associated subpopulations by integrating bulk and single-cell sequencing data. *Nature biotechnology* **40**, 527-538 (2022).
7. Zeng, Z. et al. OmicVerse: a framework for bridging and deepening insights across bulk and single-cell sequencing. *Nature Communications* **15**, 5983 (2024).
8. Stephenson, E. et al. Single-cell multi-omics analysis of the immune response in COVID-19. *Nature medicine* **27**, 904-916 (2021).
9. Cesta, M.C. et al. Neutrophil activation and neutrophil extracellular traps (NETs) in COVID-19 ARDS and immunothrombosis. *European Journal of Immunology* **53**, 2250010 (2023).
10. Yang, L. et al. The signal pathways and treatment of cytokine storm in COVID-19. *Signal transduction and targeted therapy* **6**, 255 (2021).
11. Acharya, D., Liu, G. & Gack, M.U. Dysregulation of type I interferon responses in COVID-19. *Nature Reviews Immunology* **20**, 397-398 (2020).
12. V'kovski, P., Kratzel, A., Steiner, S., Stalder, H. & Thiel, V. Coronavirus biology and replication: implications for SARS-CoV-2. *Nature Reviews Microbiology* **19**, 155-170 (2021).
13. Zappia, L., Phipson, B. & Oshlack, A. Splatter: simulation of single-cell RNA sequencing data. *Genome Biology* **18**, 174 (2017).

14. Moffitt, J.R. et al. Molecular, spatial, and functional single-cell profiling of the hypothalamic preoptic region. *Science* **362**, eaau5324 (2018).
15. Li, Y. et al. in Proceedings of the AAAI conference on artificial intelligence, Vol. 35 8547-8555 (2021).
16. Samala, R.K. et al. Breast cancer diagnosis in digital breast tomosynthesis: effects of training sample size on multi-stage transfer learning using deep neural nets. *IEEE transactions on medical imaging* **38**, 686-696 (2018).

We sincerely thank the reviewers and editorial team for their insightful feedback on our manuscript

"scCobra allows contrastive cell embedding learning with domain adaptation for single-cell data integration and harmonization" (ID: COMMSBIO-24-4123A. We are delighted to see that Reviewer #1 and Reviewer #3 are fully satisfied with our revision, and we appreciate their recognition of our comprehensive efforts. In this round, we have focused on refining the manuscript further by incorporating Reviewer #2's minor comments. Enclosed are our point-by-point responses to the concerns raised and the amendments applied to the manuscript (all changes were tracked in blue).

Reviewer #2

The manuscript has improved substantially following the authors' comprehensive revisions. However, there remain a few points that require further clarification:

C1. Please enlarge the figure legends and text size in both the main figures and supplementary figures to enhance readability.

Response: We sincerely appreciate your constructive comments, which have significantly contributed to improving the clarity and quality of our manuscript. We also thank you for recognizing the comprehensiveness of our revisions. Additionally, we appreciate your emphasis on enhancing figure readability. As per your suggestion, we have systematically enlarged text and legends in the following figures: Fig. 2c, d, e, f; Fig. 3d; Fig. 5b, c; Fig. S2b; and Fig. S9. These adjustments ensure improved visual clarity while maintaining the original data resolution. We have carefully reviewed all figures to meet high readability standards and ensure their accessibility for all readers.

C2. It is unclear whether a higher or lower value of the overcorrection score is considered optimal in supplementary fig.6. Please specify any criteria or thresholds used to identify the best results.

Response: We appreciate your insightful comment. The over-correction score quantifies the risk of over-correction, where higher values indicate greater over-correction, while lower values correspond to a lower over-correction risk. The theoretical highest score corresponds to no correction at all (0). As per your suggestion, we have updated the figure captions for Fig. 3c and Fig. S6e, as well as the corresponding text in the manuscript, to explicitly clarify that lower scores represent more optimal outcomes. We believe these refinements ensure clarity and eliminate any ambiguity.

C3. The first plot in Figure 3b is labeled "IR27" instead of "IL27." Kindly double-check and correct any similar errors throughout the manuscript.

Response: We sincerely appreciate your careful review and for bringing this oversight to our attention. We have corrected the labeling in Figure 3b to ensure the correct marker, "IL27," is used. Additionally, we have systematically reviewed the entire manuscript to verify the accuracy of all marker labels, and we confirm that no similar errors remain.